# A lineage tree-based hidden Markov model quantifies cellular heterogeneity and plasticity

Farnaz Mohammadi [1], Shakthi Visagan [1], Sean M. Gross[2], Luka Karginov[3], J. C. Lagarde[1], Laura M. Heiser [2] & Aaron S. Meyer [1,4,5,6 ✉]

Individual cells can assume a variety of molecular and phenotypic states and recent studies indicate that cells can rapidly adapt in response to therapeutic stress. Such phenotypic plasticity may confer resistance, but also presents opportunities to identify molecular programs that could be targeted for therapeutic benefit. Approaches to quantify tumor-drug responses typically focus on snapshot, population-level measurements. While informative, these methods lack lineage and temporal information, which are particularly critical for understanding dynamic processes such as cell state switching. As new technologies have become available to measure lineage relationships, modeling approaches will be needed to identify the forms of cell-to-cell heterogeneity present in these data. Here we apply a lineage tree-based adaptation of a hidden Markov model that employs single cell lineages as input to learn the characteristic patterns of phenotypic heterogeneity and state transitions. In benchmarking studies, we demonstrated that the model successfully classifies cells within experimentally-tractable dataset sizes. As an application, we analyzed experimental measurements in cancer and non-cancer cell populations under various treatments. We find evidence of multiple phenotypically distinct states, with considerable heterogeneity and unique drug responses. In total, this framework allows for the flexible modeling of single cell heterogeneity across lineages to quantify, understand, and control cell state switching.

[1] Department of Bioengineering, University of California, Los Angeles, CA, USA. [2] Department of Biomedical Engineering, Oregon Health & Science University, Portland, OR, USA. [3] Department of Bioengineering, University of Illinois, Urbana Champaign, IL, USA. [4] Department of Bioinformatics, University of California, Los Angeles, CA, USA. [5] Jonsson Comprehensive Cancer Center, University of California, Los Angeles, CA, USA. [6] Eli and Edythe Broad Center of Regenerative Medicine and Stem Cell Research, University of California, Los Angeles, CA, USA. ✉email: ameyer@ucla.edu

Chemotherapy and targeted therapies selectively eliminate fast-proliferating or oncogene-addicted cells and are among the primary treatments for cancer. However, long-term therapeutic efficacy is inevitably limited by widespread intratumoral heterogeneity[1,2]. Cell-to-cell variability in drug response can originate from cell-intrinsic factors—such as genomic alterations, epigenetic mechanisms like changes in chromatin state[3], and variable protein levels[4,5]—or cell-extrinsic factors such as spatial variability in the surrounding vasculature and environmental stressors[6–8]. Moreover, cell plasticity, where cells adopt new characteristics such as those of other cell types, is observed in cancer cells, and can affect their sensitivity to therapy[9].

Large-scale profiling studies can find molecular features that associate with drug response using population-level samples[10,11]. These associations, while valuable, can miss the contribution of cell-to-cell heterogeneity, and especially stochastic changes in individual cell states that compound to effects on overall tumor drug response[3,12,13]. The most common methods for quantifying drug response are metrics of tumor cell population expansion or contraction[14–17]. Recent research has made efforts to track phenotypic measurements of fitness at the single-cell level[18,19], however, even single-cell measurements are typically performed with snapshots that subsequently miss the role of individual cells in the overall population response[20]. Though population heterogeneity is usually defined through molecular measurements, studies that have explicitly linked molecular and phenotypic variation have been able to identify mechanisms that underlie cell-to-cell variation that would otherwise remain hidden[21], and studies starting with phenotypic analysis have generally found that phenotypic variability arises from a small number of molecular factors leading to the phenotypic variation[4,22,23].

Measurements accompanied by lineage relationships are uniquely valuable for studying inherited phenotypes within families of individuals. This value is evident in linkage studies wherein relatives are used to identify or refine the genetic determinants of disease[24–26]. Notably, linkage studies can identify genetic determinants with greater power than even large association studies because relatives essentially serve as internal controls[27]. Linkage studies also start with the phenotype of individuals, rather than grouping based on molecular differences, ensuring discoveries are phenotypically consequential. While the inherited factors are different between cells (e.g., proteins, RNA) and people (DNA), such approaches are likely to be similarly useful with populations of cells. Recently, constructing phylogenic trees of cancer cells using lineage tracing and single-cell sequencing has helped to characterize the directionality of metastatic seeding, though these methods are limited to tracking slow processes such as mutational differences[28]. Lineage-resolved data has also demonstrated value in uncovering cell-to-cell heterogeneity due to transient differences outside of cancer[22,23]. Therefore, tools to analyze and explore these data will be critical to uncovering new forms and sources of cell-to-cell variation.

Hidden Markov models (HMMs) provide an efficient strategy to infer discrete states from measurements when a series of co-dependent observations are made. An example of this is their widespread use in time series analysis, where each measurement is dependent on those that came before[29,30]. Recognizing this co-dependence allows HMMs to make accurate inferences even in the presence of extremely noisy measurements since each neighboring measurement provides accumulating evidence[31]. These models derive their relative simplicity by assuming a Markov process, meaning that the current behavior of a system can be assumed to be independent of its earlier history should its current state be known. This assumption naturally applies in many contexts. In the case of cells, this assumption aptly captures cell inheritance because daughter cells inherit both molecular signals and their environment from their predecessor. Indeed, several recent examples of cell-to-cell inheritance mechanisms can be represented as a Markov process through linear chains or cycles of states[12,22,23]. HMMs have been adapted to lineage trees (tHMMs) so that each measurement across the tree can similarly provide accumulating evidence for a prediction. Just like with time-series data, these models can provide very accurate predictions despite noisy measurements and limited information by recognizing the co-dependence between related measurements[32,33]. tHMMs have been used in a multitude of applications, from image classification to comparative genomics[34,35]. These models have been fit to lineages collected from stem cells and bacteria colonies, but have always required custom implementations[36,37]. Improvements in cell tracking and high-throughput imaging promise to make tHMM models valuable techniques for studying the plasticity of heterogeneous cell populations. However, widespread use of these models still depends on more easily usable implementations, examples of successful tHMM-based discoveries, and standards for experimental application.

Here, we develop an extensible implementation of tHMMs with a defined interface for integrating diverse types of measurements on cell lineage trees. This model allows us to quantify the dynamics and phenotypic features of drug response heterogeneity. We leveraged information about the relationships between cells to analyze the cell cycle responses of populations of breast cancer cells to a panel of therapies, and how normal breast cells respond to growth factor treatment. Single-cell measurements of the cell cycle revealed extensive variation not captured by population-level measurement. Using the tHMM model, we inferred the number of phenotypically distinct subpopulations, the characteristics of those subpopulations, the transition probabilities from one state to another, and each cell's expected state. We also confirmed that the tHMM model could use patterns of inheritance to predict cell behavior. This work, therefore, provides a flexible phenotype-driven route to discovering cell-to-cell variation in drug response, demonstrates an overall strategy for quantifying the dynamics of cell heterogeneity, and implements a very general software tool for the widespread use of tHMM models.

## Results

**Lineage information provides unique information about the source and structure of cell-to-cell heterogeneity.** Single cells grow and then divide into two daughter cells, eventually forming a binary genealogical tree, also known as a lineage tree. We collected single-cell measurements in the form of lineage trees to track these relationships. The life cycle of each cell before division includes G1, S, G2, and M phases that must pass one after another. To illustrate the unique value of lineage measurements in analyzing intra-tumoral and drug response heterogeneity, we collected cell fates (whether cells ultimately divide or die) alongside either cell lifetimes (MCF10A) or individual cycle phase durations (AU565). Two random subsets of the tracked lineages of the breast cancer cell line AU565 are plotted in Fig. 1a. The single cell lineages reveal striking variation in cell cycle phase durations and cell division dynamics despite coming from the same sample. Population-level measurements would be unable to identify this difference as the starting and ending cell numbers are the same. Measurements that record or reflect the history of cells (e.g., CFSE staining, Luria-Delbruck experiment) can help to identify these variations within cell populations but must make assumptions about the dynamics of heterogeneity[13,23]. Lineage measurements, by contrast, provide sufficiently rich temporal

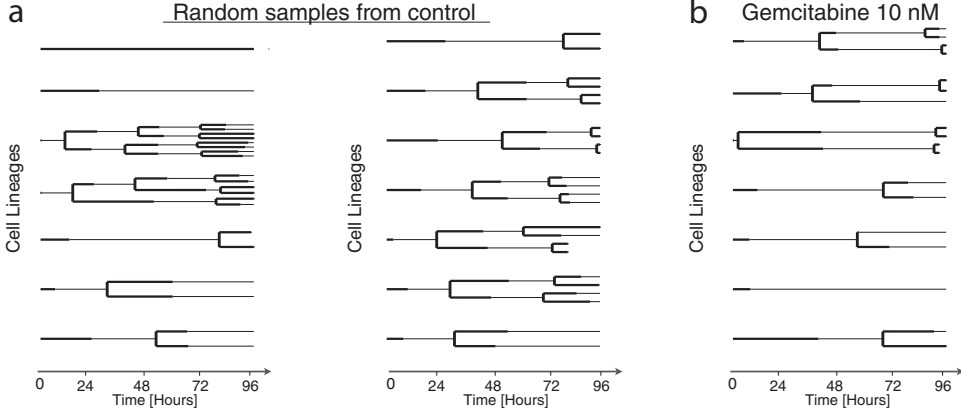

**Fig. 1 Total cell number is insufficient to distinguish the structure of heterogeneous populations. a** Randomly sampled lineages of untreated AU565 cells from the same replicate and experiment. **b** Randomly sampled lineages of AU565 cells treated with 5 nM gemcitabine from a single replicate and experiment. Each line indicates the lifetime of one cell. A line branching into two lines indicates cell division. The G1 and S/G2 phase durations are indicated by solid thick and thin lines, respectively.

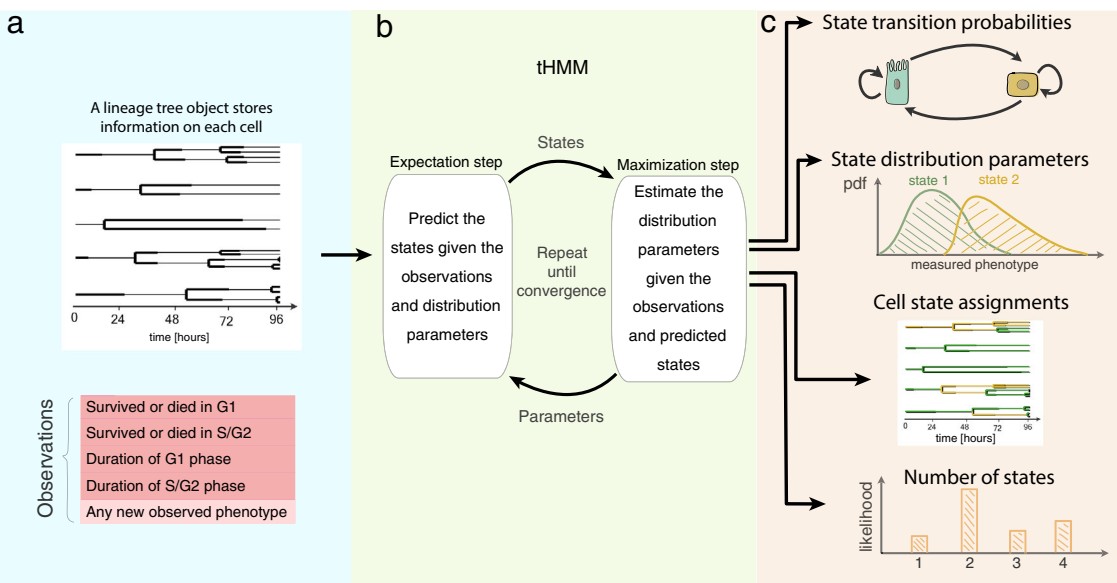

**Fig. 2 The tHMM model. a** Input data takes the form of single-cell measurements across time, where the lineage relationship between cells is known. **b** The fitting process includes expectation and maximization steps, where model parameters are iteratively updated until convergence. **c** Output predictions of the model after fitting including the tree of inferred cell states, probabilities of transition between each state, starting abundance of each cell state, and distributions that describe the behavior of cells within each state. The model likelihood can be used to estimate the number of distinguishable cell states.

information to quantify the specific structure of the phenotypic heterogeneity.

As further exploration of the cell tracking data, we randomly sampled lineages from gemcitabine-treated AU565 cells (Fig. 1b). Gemcitabine is a chemotherapy agent that disrupts DNA replication and results in the extension of and apoptosis in S phase[38]. We found that the S/G2 phase lengths in treated cells were noticeably extended compared to untreated cells, slowing population growth. There was generally striking variation between lineages of a single condition, including anywhere from zero to three cell divisions, but tightly shared behavior among cells and their relatives in each lineage. These observations demonstrate some of the unique advantages of collecting lineage-based measurements.

**A lineage tree-based hidden Markov model infers the state of cells given measurements on lineage trees**. Given the unique insights that single-cell measurements on lineage trees can

provide, we implemented a strategy for classifying cells based on their phenotype and lineage relationships. We used a tree-based hidden Markov model (tHMM) to fit a set of measurements made across a lineage tree (Fig. 2a). Like a typical hidden Markov model, a tHMM can infer the hidden discrete "states" of cells given a series of measurements where a state is defined by specific phenotype distributions. The inference of these states takes place using an iterative strategy wherein the states of each cell are predicted by the phenotype of both the cell and its relatives in a lineage (expectation step), and then each distribution of phenotypes is fit to match the cells within that state (maximization step) (Fig. 2b). This expectation-maximization (EM) process repeats until convergence.

After fitting, the model can provide a variety of information (Fig. 2c). First, it infers the starting and transition probabilities of each state. Second, the distribution of cells' phenotypes in each state are estimated and can be compared to distinguish how cells of each state behave. For instance, if we use the growth rates of cells as their

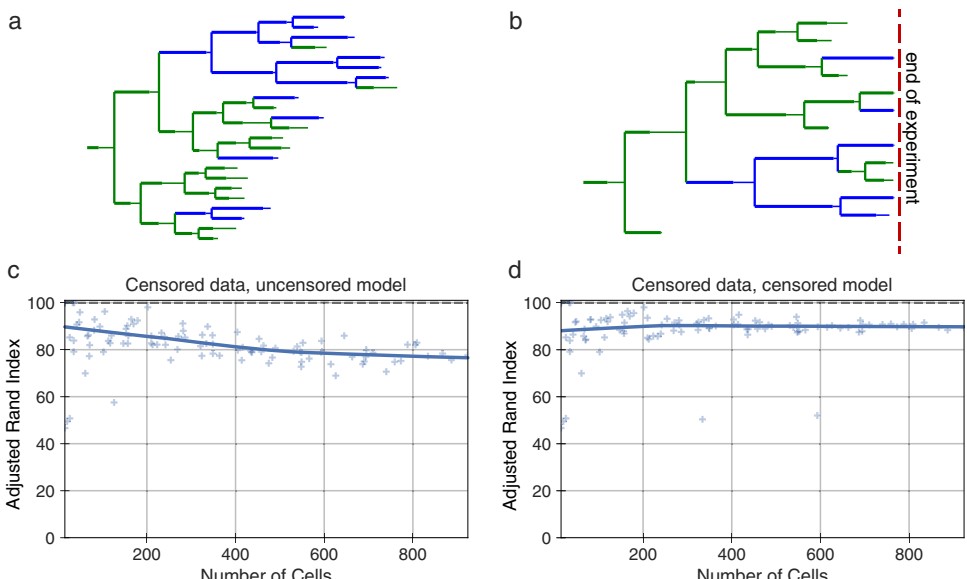

**Fig. 3 Experiments of finite time necessitate data censorship corrections. a** An example synthetic, uncensored two-state lineage. **b** An example synthetic, censored two-state lineage. Cells in state 0 and 1 are shown in green and blue, respectively. **c** State assignment accuracy with censored lineages using an uncorrected model. **d** State assignment accuracy with censored lineages using the corrected model. Each scatter point represents the state assignment accuracy of the model when fit to a lineage with the indicated number of cells. The solid lines show the Lowess trendline of the individual run accuracies. 100 trials are plotted. The accuracy of state assignment is measured by the adjusted rand index, a similarity measure that can serve as an accuracy for unlabeled clustering problems.

phenotype, we may observe a subpopulation of cells with shorter times to division, and another with longer times. Moreover, the state of each individual cell can be predicted from the fit data or new measurements. Finally, the model provides a likelihood of each cell's observations and therefore the data overall. This last quantity can be used, for example, to estimate the number of distinguishable cell states. When implementing these processes, we ensured that a cell's measurements were defined through a modular interface, allowing many other forms of data to be easily integrated, such as cell morphology or molecular measurements.

**Experiments of finite time necessitate corrections for experimental censorship**. Modeling the duration of each cell's lifetime is complicated by the influence of experimental parameters. Specifically, cells measured at the beginning or end of an experiment persist beyond the experiment's duration and so, while we observe these cells, we do not know their exact lifetimes. Data censorship occurs when a measurement is systematically affected by an undesired influence. For instance, in our case, phase durations are censored because the experiment started after cells had already begun their initial cell cycle phase or the experiment ended before they had completed their last phase. Previously, this has been addressed by removing incompletely observed cells[22]. However, doing so results in a systematic bias, where longer-lived cells are preferentially eliminated. On the other hand, ignoring the truncation of these values also creates bias by creating an upper bound on the cells' lifetimes (Fig. 3b, c).

To correct for this effect in our model, we marked cells that encountered the start or end bounds of the experiment. When estimating the properties of these cells' lifetime we instead used a censored estimator or the survival function of the distribution[39]. Because the labels are interchangeable in our classification, we used the adjusted rand index[40], a similarity measure that can serve as an accuracy measure for clustering results. Using synthetic data, we verified that this correction resulted in accurate phenotype estimations (Fig. 3d, Supplementary Figs. 3, 10). Thus, accounting for cells that outlive the bounds of the experiment

through a censored estimator removes the contribution of this experimental confounder.

**Synthetic lineage benchmarks show a tHMM can accurately infer population behavior**. To evaluate how accurately a tHMM model could infer the behavior of multi-state cell populations, we used synthetic populations of cells in a wide variety of configurations, such as various populations sizes, numbers of states, and abundance of the states. In each case, we determined that the tHMM model could accurately infer the hidden states and parameters of a population given at least 100 cells. This synthetic data included uncensored (Supplementary Figs. 1, 2, 8, 9; Supplementary Tables 1, 2) or censored (Supplementary Figs. 4, 10, 3, 15; Supplementary Tables 1–3) situations. Synthetic data were created by lengthening the simulated experiment time, in effect creating deeper lineages, or by increasing the number of initial cells to have a greater number of lineages, increasing the experiment's breadth. In addition to varying the number of cells in a population, we benchmarked populations with varied cell state percentages (Supplementary Figs. 4, 5) and varied the degree of phenotypic differences between states (Supplementary Figs. 6, 7; Fig. 5). This benchmarking consistently showed that the tHMM model would provide accurate results across a range of circumstances, and generally provided accurate results with datasets consisting of at least 10 lineages, 100 cells overall, and 10 cells from each state.

More specifically, one of the benchmarking studies we performed was with data matching our measurements of AU565, where G1 and S/G2 phase durations were represented by a gamma distribution, and their corresponding cell fate represented by a Bernoulli distribution (Fig. 4). The choice of the gamma distribution for cell cycle phase was inspired by a previous study[41] and verified by evaluating a variety of distributions; the gamma distribution fit the cell lifetime data best. Although the tHMM model was fit with no information about the true underlying parameters of the simulated cells, it distinguished the pre-assigned two underlying cell states'

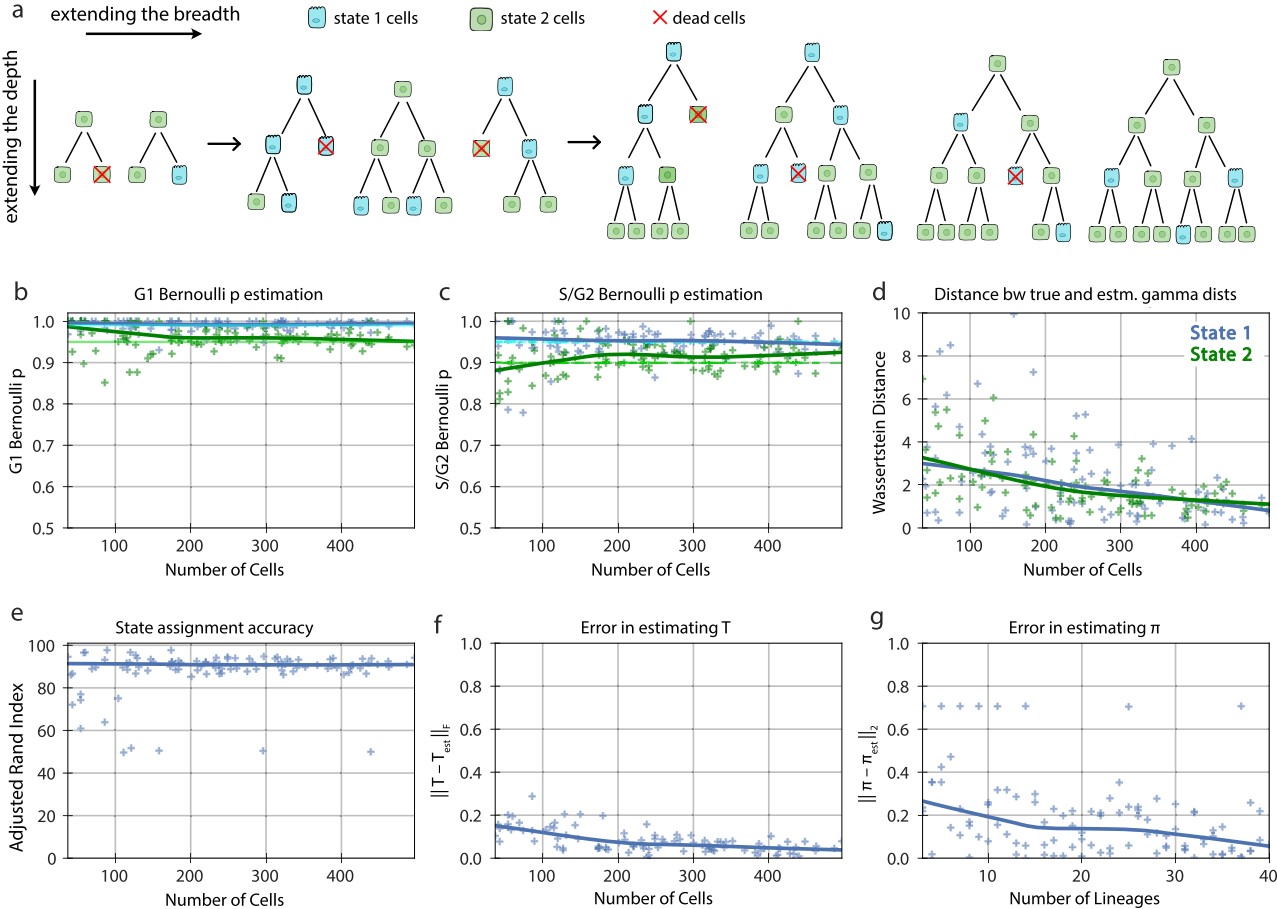

**Fig. 4 Model performance on censored lineages of two states with increasing breadth and depth. a** Synthetic two-state populations of increasing breadth (increasing number of initial cells and therefore lineages) and of increasing depth (increasing experiment time and therefore more cells in each lineage) are analyzed. The states are shown as green and blue colors. Red indicates cell death. **b, c** The accuracy of estimating the Bernoulli parameters for G1 and S/G2 phase, respectively. Each point in the scatter plots represents the inferred value for a model evaluation trial with the number of cells shown in the x-axis. The dark solid lines are the Lowess trendline across the individual trials. The light green and light blue lines show the true value of the parameters. **d** The distance between the true and estimated gamma distributions associated with phase lengths for the two states. **e** The state assignment accuracy. **f** The errors in the estimated and transition rate matrices. **g** The initial probability vector. Note that the Wasserstein distance between the true and estimated distributions for each state is much lower than the distance between two distributions that are quite similar (Fig. 5b). 100 simulation trials are plotted.

phenotypes (Fig. 4b–d) and member cells with >95% accuracy (Fig. 4e). The Wasserstein distance metric was used to quantify the difference between the true and estimated cell cycle phase duration distributions to show the accuracy of parameter estimation (Fig. 4d). On the population level, the difference between the true and estimated transition probabilities, as calculated by the sum of squared difference, was less than 0.1 for 100 cells or more. Starting probabilities were compared to their corresponding true values using the Euclidean distance and showed less than a 0.2 error for populations with 10 lineages or more (Fig. 4f, g). Thus, we are confident that with similar experimental data, we should derive accurate results.

**Lineage information improves cell state identification with heritable phenotypes.** Cells of even very distinct molecular states can have partly overlapping phenotypes due to non-heritable variation. Therefore, we sought to evaluate how different two states need to be for us to accurately identify them as distinct (Fig. 5a). We varied the G1 phase duration of two states from identical to very distinct (Fig. 5b) and quantified the state assignment accuracy of our model (Fig. 5c). While the phenotypic observation of a given state had to be different for our model to accurately assign cells,

even moderately overlapping phenotypes (Wasserstein distance of ~20) could be distinguished by using the lineage relationships of cells. As a baseline comparison, we analytically identified the optimal classifier in the absence of lineage information (see Methods). The tHMM consistently outperformed this approach (Fig. 5c). The model performance in censored and uncensored populations was similar (Supplementary Figs. 6, 7). This shows that lineage relationships can be used to identify cell states with partially overlapping phenotypes more accurately.

**Likelihood-based model selection can effectively identify the number of distinct states.** One does not usually know the number of distinct cell states within a population. Further, the number of distinct states may vary depending upon the environmental context of the cells, particularly for phenotypic measurements[42,43]. To test whether we could infer the number of phenotypically distinct states, we performed model selection using the Bayesian information criterion (BIC) while varying the number of states in synthetic data (Fig. 6). We normalized the BIC values such that zero corresponds to the state with the highest likelihood. The synthetic populations included approximately 250 to 650 cells with known cell phase fate and phase lengths (Supplementary Table 3). The

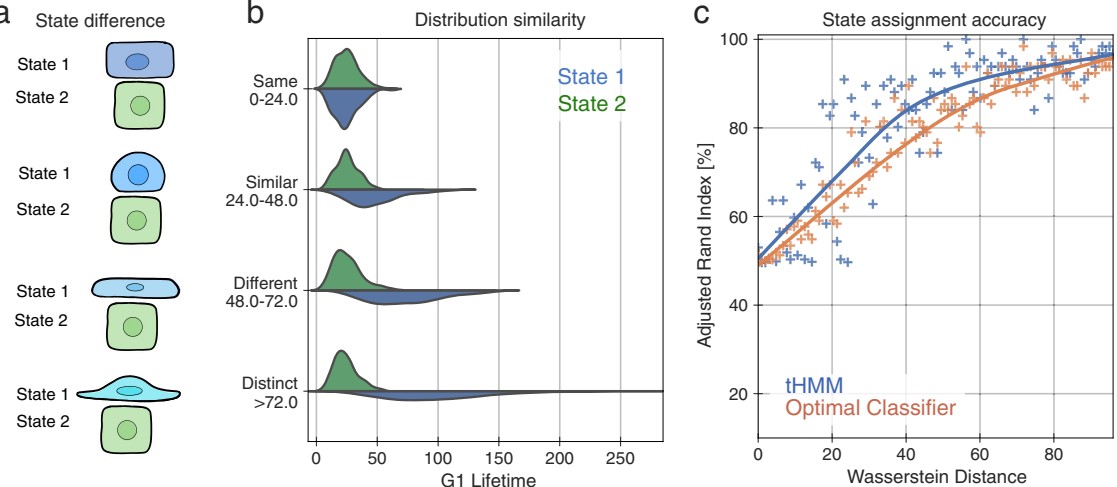

**Fig. 5 Model performance versus the difference between states. a** Cartoon of how two states can vary in their phenotypic similarity, in a synthetic population of two states. On the top, cells might be virtually indistinguishable (here based on shape). On the bottom, they might be so different that looking at one cell is sufficient to identify its state. **b** The distribution of G1 duration is varied in state 1 (blue) while the other state is kept constant. **c** State assignment accuracy versus the Wasserstein distance between state phenotypes. Each point represents the accuracy of state assignment for a lineage created by a set of parameters that yield the shown Wasserstein distance between the two-state distributions. 100 simulation trials are plotted. Either the tHMM model (blue) or an optimal classifier without lineage information (orange) was used. The solid lines show a Lowess trendline of the model accuracy.

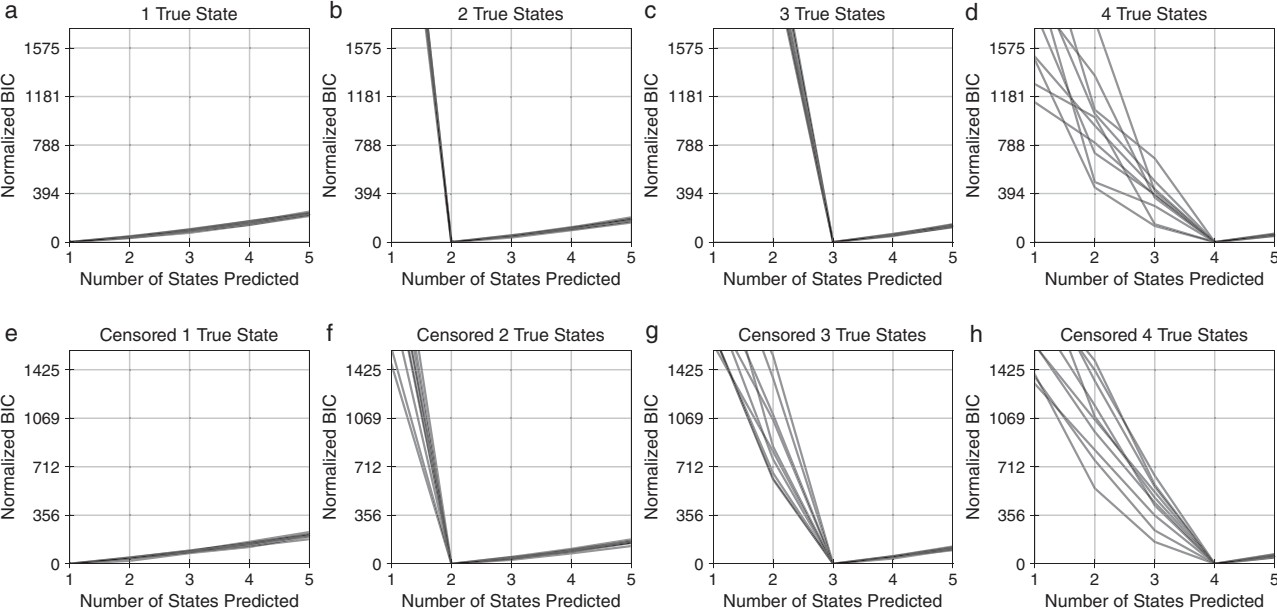

**Fig. 6 Model selection effectively identifies the number of distinct states in synthetic data. a–d** Model BIC for synthetic uncensored lineages with 1–4 true states. **e–h** Model BIC for synthetic censored lineages with 1–4 true states. BIC values are normalized such that the optimum is equal to 0. The minimum BIC value corresponds to the predicted number of states in each repetition. 10 trials plotted.

inferred number of cell states was consistently correct for both uncensored (Fig. 6a–d) and censored lineages (Fig. 6e–h). This indicated that model selection can help to identify the appropriate number of cell states for a set of measurements.

**tHMM infers several distinct subpopulations in experimental drug response data.** As an application of our model, we used phenotypic measurements from two cell lines. With the first, AU565, we measured of the G1 and S/G2 phase durations and terminal cell fates of cells in a control condition and when treated with 3 concentrations of gemcitabine or lapatinib. For the second, MCF10A, we measured the overall cell lifetimes and terminal

fates of cells treated with PBS or single concentrations of the growth factors EGF, HGF, or OSM. Cells were imaged every 30 minutes and then tracked over time to assemble lineage relationships. The lapatinib and gemcitabine-treated AU565 populations (including control) contained a total of 5290 and 4537 cells, respectively. The MCF10A population contained 1306 cells. Lineages included 1–5 generations of cells. The model was fit to each experiment's data across all conditions, enforcing that the initial and transition probabilities are shared across concentrations but allowing the phenotype distributions to vary. We enforced a unidirectional phenotypic shift with drug concentration in AU565 cells, reflecting the expectation of a dose-response effect on cell phenotype within each state. The cell fate

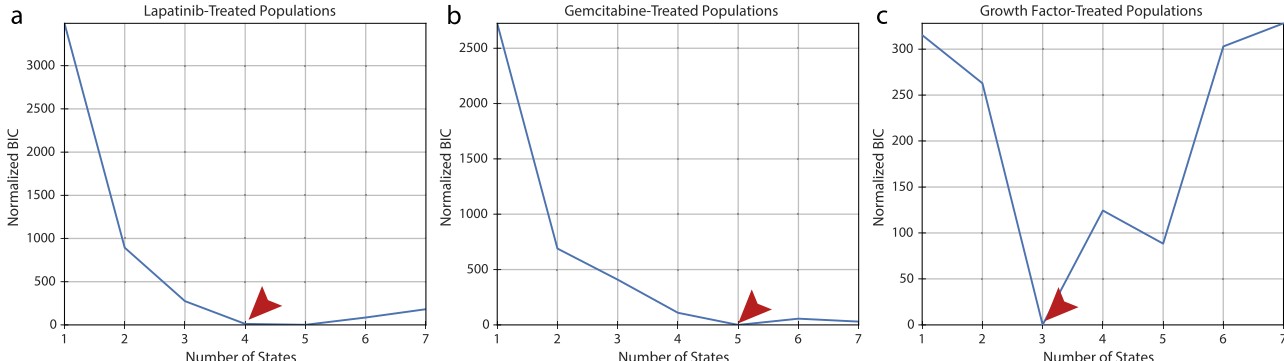

**Fig. 7 BIC-based model selection infers the number of phenotypically distinct states.** Normalized BIC values for (**a**) AU565 cells in control and treated with 5 nM, 25 nM, and 250 nM of lapatinib; (**b**) AU565 cells in control and treated with 5 nM, 10 nM, and 30 nM of gemcitabine; and (**c**) MCF10A cells treated with PBS, 10 ng/ml EGF, 40 ng/ml HGF, and 10 ng/ml OSM. The BIC values for all conditions were normalized such that the minimum value was zero. The arrows in (**a–c**) point to the optimal number of states.

parameters were estimated without constraints. We assumed the number of states is shared across drug concentrations in AU565 cells and across growth factor treatments in MCF10A cells. To determine the number of cell states, we compared models of 1–7 states using the BIC, where the lowest BIC value across numbers of states indicates the most optimal model correcting for complexity (Fig. 7a–c). The data for each compound indicated the presence of multiple inherited states.

To verify the model's predictive ability, we additionally implemented a cross-validation scheme for the lineage data. Briefly, roughly 20% of the cells were chosen at random and then masked from the fitting process. The model parameters were estimated using only the unmasked cells, though all cells received state assignments through use of their relatives. At the end, the log-likelihood of the masked cells' observations were evaluated using the fit model. We tested this cross-validation approach by creating synthetic cell populations of 2–5 true states with conditions matching the experimental data. For each scenario, we were able to identify the correct number of states based on which gave the highest log-likelihood (Supplementary Fig. 16a–c, f, g, Supplementary Table 3). Cross-validating the experimental data again confirmed the 4 and 5 phenotypic states within the lapatinib and gemcitabine data, respectively (Supplementary Fig. 16d, e). It also directly demonstrated that the inclusion of multiple states enables the tHMM model to predict unseen data, and that this prediction is dependent on inheritance; a no-inheritance model, in which all transitions were equally likely, performed relatively poorly (Supplementary Fig. 16d, e).

**Lapatinib response is defined by both stable and inter-converting states.** We fit the lapatinib-treated data to the model with 4 states based on our BIC-based model selection, confirmed by cross-validation (Fig. 7a, Supplementary Fig. 16f). Fitting revealed states of widely varying persistence over generations, from less than a 0.01 probability of remaining in state 2 to a 0.94 probability of remaining in states 1 and 3 (Fig. 8a). Interestingly, states 2 and 4 formed a cycle wherein the most probable transition was between the two (Fig. 8a, Supplementary Fig. 11).

Examining the phenotypes of each state revealed distinct drug responses. Lapatinib is an EGFR/HER2 inhibitor that induces cell cycle arrest in G1 phase[44]. Every state displayed a dose-dependent increase in G1 phase lifetime with lapatinib treatment, and G1 effects were more pronounced as compared to those involving S/G2 (Fig. 8b–i, Supplementary Fig. 11). While the probability of survival at the end of the cell cycle phase decreased at higher concentrations, very few cell death events were observed (Fig. 8h, i, Supplementary Fig. 11). Consequently, the chances of cell death likely have high

uncertainty at higher concentrations of lapatinib. States 2 and 4 were highly arrested in both G1 and S/G2 phase; in contrast, states 1 and 3 experienced little arrest in G1 and no arrest in S/G2 phase (Fig. 8f, g). Thus, cell states seemed to be primarily distinguished based on the degree of lapatinib response. The cycle between states 2 and 4 seems to reflect the observation that cells more highly arrested in G1 than G2/S give rise to cells that spend longer in G2/S than G1, and vice versa (Fig. 8, Supplementary Fig. 11).

**Gemcitabine-treated populations are clustered into phase-specific responses.** Gemcitabine is a chemotherapy agent that induces cell cycle arrest and apoptosis in S phase by disrupting DNA repair. The AU565 cells were treated with 5, 10, and 30 nM of gemcitabine; model selection, confirmed by cross-validation, inferred 5 states in the population (Fig. 7b, Supplementary Fig. 16a/g). Examining the 5-state fit revealed relatively stable states 1, 3, and 4 (Fig. 9a–e, Supplementary Fig. 12). States 2 and 5 formed a cycle with high rates of interconversion.

Gemcitabine modulated both G1 and S/G2 cell cycle phases and these effects were variable across the five identified states (Fig. 9f–i). State 5 showed S/G2-specific arrest and always resulted in cell death at the highest concentration (Fig. 9g, i, Supplementary Fig. 12). Meanwhile, cells in state 4 grew almost normally, with some cell death in G1 at the highest concentration (Fig. 9f–i). At the highest concentration, state 3 represents the cells arrested at S/G2 that have not divided even once, and state 5 is the representative of almost all cells undergoing cell death at S/G2 (Fig. 8i, Supplementary Fig. 12).

Lastly, we wished to explore whether the phenotypic heterogeneity we observed was limited to cancer cells or cytotoxic drug treatment. To determine this, we tracked non-tumorigenic MCF10A breast cells. These cells are normally grown in the presence of epidermal growth factor (EGF); we compared this condition to growth factor withdrawal (PBS) or rescue with hepatocyte growth factor (HGF) or oncostatin M (OSM)[45]. Each growth factor consistently promoted proliferation on a population level compared to the PBS control, though with considerable inter- and intra-lineage variation (Supplementary Fig. 13). BIC-based model selection inferred the presence of 3 distinct states (Fig. 7c). Inspecting the model revealed generally more dynamic transitions between states as compared to the AU565 experiments (Supplementary Fig. 14). Due to the lack of growth factors, most cells arrested in the PBS condition; few observations of either division or death events is the reason for the division probability being 0.5 (Supplementary Fig. 14g). State 1 was distinct in being relatively less responsive to HGF and OSM treatments (Supplementary Fig. 14a/f), while state 1 displayed

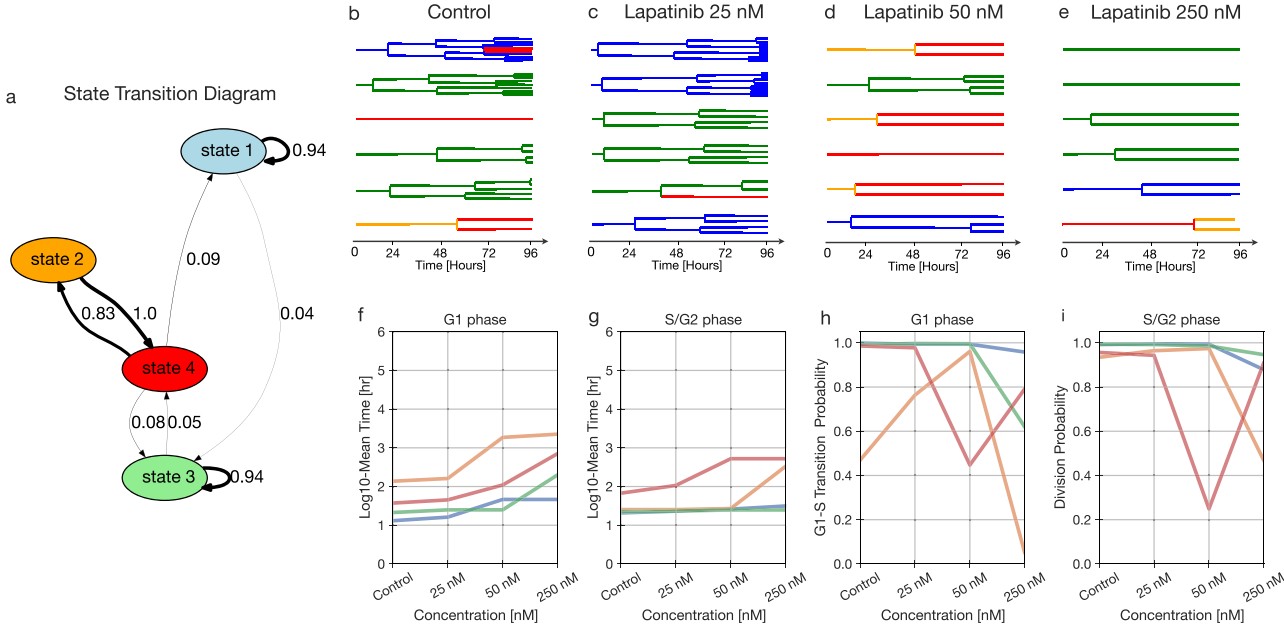

**Fig. 8 Lapatinib response is defined by phenotypically-distinct stable and interconverting states. a** State transition graph showing the probability of state transitions among the predicted states. Transitions with less than a 0.03 probability have been removed. **b–e** A sample of lineage trees after fitting the model and state assignment (control, 25 nM, 50 nM, and 250 nM). **f–g** The $\log_{10}$ of fit mean time of G1 and S/G2 phase durations for different concentrations. **h**, **i** The Bernoulli parameter, indicating the probability of G1-to-S phase transition versus cell death (**h**), and the probability of division versus cell death (**i**) for each concentration.

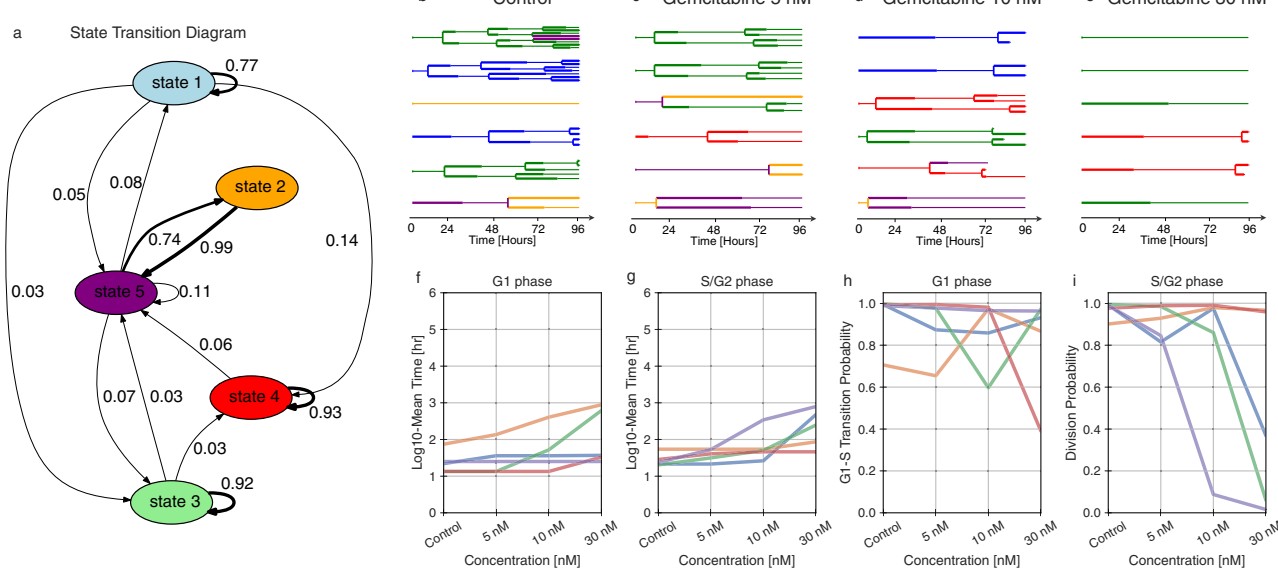

**Fig. 9 State-specific inferences of the gemcitabine-treated data. a** State transition graph showing the probability of state transitions among the predicted states. The transitions with less than a 0.03 probability have been removed. **b–e** A sample of lineage trees after fitting the model and state assignment (control, 5 nM, 10 nM, and 30 nM). **f**, **g** The $\log_{10}$ fit mean time of G1 and S/G2 phase durations for different concentrations. **h**, **i** The Bernoulli parameter, indicating the probability of G1-to-S phase transition versus cell death (**h**), and the probability of division versus cell death (**i**) for each concentration.

higher rates of cell death overall (Supplementary Fig. 14a/g). In total, the tHMM model was effective in identifying subsets of cells with divergent phenotypic responses to drug treatment alongside the relationships between cells in the population.

## Discussion
Heterogeneity and plasticity in cancer cells enables them to adapt in response to therapy. Even in the absence of genetic mutations,

other heritable variation serves as a substrate for selection[46,47]. In this paper, we introduced a tree-based hidden Markov model that clusters single cells from heterogeneous populations based solely on their phenotypic traits and relationships. Model benchmarking showed that it can provide accurate results using feasible experimental designs. Of particular importance, the tHMM could recognize subpopulations even at lower frequencies (Supplementary Figs. 6, 7). Comparing the model to more standard clustering, the tHMM showed that lineage information helps to

identify cell states more accurately (Fig. 5). Using cross-validation, we were able to show that accounting for cell inheritance allowed the model to accurately predict unseen observations (Supplementary Fig. 16). Several critical advancements in the current work are a modular interface for using tHMM models with various phenotypes (Fig. 2), proper censorship handling (Fig. 3), strategies for model evaluation (Fig. 7, Supplementary Fig. 16), and demonstrating that such a model can be applied to study cancer heterogeneity at baseline and in response to perturbation.

We used single-cell lineage tracking data of AU565 cancer cells treated with lapatinib and gemcitabine as a demonstration of the model. G1 and S/G2 cell cycle phase durations and cell fate measurements were used as relevant cell phenotypes to quantify the anti-cancer effects of these drugs. We were able to identify 4 and 5 distinct subpopulations within the lapatinib and gemcitabine-treated data, respectively (Fig. 7). The phenotypic features of each state were quantified in parallel (Figs. 8, 9). Lapatinib is known to inhibit cell proliferation by inhibiting Akt/mTOR pathway activity, which is a key regulator of G1 phase progression[48]. Similarly, our analysis in the lapatinib-treated population indicated that cells, regardless of their state, experienced a prolonged G1 phase, but individual states varied in their susceptibility. In gemcitabine-treated cells, we observed that most states were highly heritable, with more varied phenotypic effects. This included cells that became arrested in S/G2 and underwent apoptosis (state 5), cells that were selectively arrested in G1 (state 1), and cells that hardly responded to drug treatment at all (state 4; Fig. 9). While gemcitabine canonically works by inducing cell arrest in G2/S, previous work has characterized its effects on G1 phase by separating the effects on both cell cycle phases[49]. They similarly identified that G1 arrest was associated with cell death, which is also evident in cells of state 3 where G1 arrest is seen alongside cell death in both phases (Fig. 9f–i). Our results would further suggest that those cells with G1 effects are molecularly and heritably distinct from those that are arrested in S/G2. MCF10A cells with growth factor-induced proliferation showed a very distinct pattern of variation, suggesting that the phenotypic cell states identified by the model reflect a confluence of cell features and treatment conditions (Supplementary Figs. 13, 14).

We present several lines of evidence supporting the accuracy of the model and the existence of heritable cell states. First, across a diverse array of benchmarking experiments, we show that the model can derive accurate conclusions from synthetic data with properties like those we observed in the experimental measurements (e.g., Fig. 4). Through an informatic model selection scheme, we find statistical evidence for the existence of multiple states (Figs. 6, 7). Examining these cell states, we find patterns consistent with the biological mechanisms of the compounds we used to alter cell proliferation (Figs. 8, 9). Reassuringly, we were able to confirm that the abundance of cell states was consistent across experimental replicates, ruling out the possibility that state differences arose from day-to-day variation between experiments. Finally, we showed that the model could more effectively predict the behavior of unseen cells with the inclusion of multiple cell states, and that this prediction is dependent on allowing inheritance between cell generations (Supplementary Fig. 16). While we have considered the use of experimental control conditions, it is important to keep in mind that the variation observed here arises both through external perturbation and natural variation within the population. Consequently, we have not been able to identify a context in which one might expect to not observe multiple states, supporting the general usefulness of our approach. While experiments in which distinct cell lines are mixed can help to validate methods in which cell relationships are inferred, such as pseudotime methods[50], the cell relationships are not modeled

here because they are explicitly known through the measured lineage relationships. Ultimately, experiments uncovering molecular markers and mechanisms of these cell states will provide the best independent validation for their biological significance.

Modeling advancements will further improve on our approach. Cells may express a continuum of, rather than discrete, phenotypic states[22]. If this is the case, a continuous latent variable model would lead to a refined view of the population-level heterogeneity. A discrete model like the one used here should, however, still provide an accurate estimate by breaking up the continuous state-space into discrete steps. Continuous latent variable models also have additional challenges in implementation and interpretation[51,52]. Careful handling of each states' phenotypic distributions might also improve the model's accuracy and power to identify distinct states. For example, the eventual fate of cells and their cell cycle durations are likely correlated which could be handled through a multivariate distribution accounting for this covariance[53]. This becomes even more important with the incorporation of other phenotypic information such as migration, cell shape, or other features, all of which are likely to be correlated to some extent.

Experimental advancements will improve the utility and accuracy of single-cell analysis using lineage information. Currently our experimental data is limited to 96 hours, covering up to five generations of cells. However, traits such as resistance may develop over more generations and longer timescales[13,54,55]. Longer data collection becomes challenging due to factors such as phototoxicity and cell stress[56]. Improved imaging modalities and experimental platforms might allow for longer tracking experiments, with reduced phototoxicity, in more physiologically representative environments such as engineered 3D extracellular matrix[57,58]. Currently, the model is agnostic as to whether the heterogeneity it identifies is pre-existing or induced by drug treatment. Collecting data in which cells are tracked before and after drug treatment, and after a wash-out, would help to link pre- and post-exposure cell phenotypes[59].

While we have identified states that represent phenotypically distinct subpopulations of cells, we currently cannot comment on the molecular factors leading to these phenotypes. Molecular barcoding has been a popular approach for identifying subpopulations of cells with genetic predispositions toward unique phenotypes, but we do not expect it would identify the same subpopulations as we do here[55]. Unlike in barcoding experiments, we do not see a bottleneck in the clonality of cells that survive treatment, and rapid interconversion between states should corrupt the relationship between ancestor phenotype and descendent molecular state[3,12]. However, we expect that single-cell molecular analyses, such as single-cell tracking tied with transcriptional profiling of the same cells at the end of the experiment, should allow us to align molecular and phenotypic states in the same populations of cells[60]. Such experiments would also provide a common baseline by which to link lineage-based phenotypic analysis and various snapshot measurements of the same cell population. In this way it should be able to pinpoint the underlying molecular mechanisms driving distinct phenotypic responses.

In total, the pipeline developed here provides a unique approach for understanding the structure of dynamic, heterogeneous tumor populations. By capturing the dynamics of state transitions, it links single-cell phenotypes to overall population behavior. Incorporating molecular measurements, and a broader set of drug interventions, will then also help to identify means of modulating state and overall population behavior. Ultimately, we expect this integrative view will help to identify treatments alone and in combination that allow for population-level control by affecting the growth of and transitions between individual cell subpopulations.

## Methods

**Experimental cell lineage data**. Stable cell line creation, drug treatments, and tracking of AU565 and MCF10A cells were performed as described in Gross et al.[61] and Gross et al.[45], respectively. Briefly, AU565 cells were co-transfected with a transposase plasmid (Addgene #34879) and a donor plasmid that drove expression of a nuclear-localized mCherry, puromycin resistance, and a fragment of HDHB fused to the clover fluorescent protein, which was used to track progression through the cell cycle[62]. Cells stably expressing the nuclear and cell cycle reporter were selected for 7 days with 0.75 μg/ml puromycin. The phase of the cells is determined based on whether the amount of fluorescence is greater within nucleus or the cytoplasm[62]. As a result, the reporter signal is invariant to changes in exposure and background. To track drug responses AU565 reporter cells were plated into 24-well plates with fluorobrite media containing 10% FBS, glutamine, and penicillin-streptomycin. 24 hours later fresh media containing escalating doses of lapatinib and gemcitabine was added. MCF10A cells were cultured in growth media (DMEM/F12, 5% horse serum, 20 ng/ml cholera toxin, 10 μg/ml insulin, and 1% Pen/Strep), grown to 50–80% confluency, and detached with 0.05% trypsin-EDTA. 7 hours after seeding 75000 cells, they were washed with PBS and the experiment media (DMEM/F12, 5% horse serum, 0.5 μg/ml hydro-cortisone, 100 ng/ml cholera toxin, and 1% Pen/Strep) was added to the 8 well-plates which was followed by 18 hours of incubation. Afterward, cells were treated with growth factors 10 ng/ml EGF, 40 ng/ml HGF, and 10 ng/ml OSM in fresh experiment media. After drug addition, plates were placed in the IncuCyte S3 and four image locations per treatment were imaged every 30 minutes. AU565 were imaged for 96 hours and MCF10A cells for 48 hours. After half the experiment times, fresh media and drugs/growth factors were added. Cell lineages from the IncuCyte images were manually tracked in FIJI[63] to record cell division, death, and the transition from G1 to S/G2 phase (in AU565). AU565 cells are non-motile and fewer than 4% of cells were within one cell length of the image boundary, ensuring minimal sampling bias from the microscopy field of view. Three biological replicates were collected and combined in the final data set. To verify that results did not reflect batch effects, we checked that state assignments were not enriched or depleted within a replicate.

**Lineage tree-based hidden Markov model**. The core assumption of a Markov chain is that the next state and current observations are only dependent on the current state. Proof of the expressions below involving cell state assignment (expectation step), including the upward recursion, downward recursion, and Viterbi algorithms, can be found in Durand[33]. All other model elements, including the emissions distribution fitting, model evaluation strategies, and censorship corrections were developed in this study.

*Basic model structure.* The initial probabilities of a cell being in state $k$ are represented by the vector $\pi$ that sums to 1:

$$\pi_k = P(z_1 = k), \quad k \in \{1, \dots, K\} \tag{1}$$

where $z$ indicates the state and $K$ is the total number of states. The probability of state $i$ transitioning to state $j$ is represented by the $K \times K$ matrix, $T$, in which each row sums to 1:

$$T_{i,j} = T(z_i \to z_j) = P(z_j|z_i), \quad i,j \in \{1, \dots, K\} \tag{2}$$

The emission likelihood matrix, $EL$, is based on the cell observations. It is defined as the probability of an observation conditioned on the cell being in a specific state:

$$EL(n,k) = P(x_n = x|z_n = k) \tag{3}$$

where $x_n$ is the observation for cell number $n$, with a total of $N$ cells in a lineage. Separate observations were assumed to be independent; for instance, cell fate is assumed to be independent from the duration of each cell phase. This facilitates calculating the likelihood of observations, because we can multiply the likelihood of all observations together for the overall likelihood.

*Assigning cell states (expectation step)*
Upward recursion: An upward-downward algorithm for calculating the probabilities in hidden Markov chains was proposed by Erphaim and Merhav[64] which suffered from underflow. This problem was originally solved by Levinson[65], where they adopted a heuristic-based scaling, and then was improved by Devijver[66] where they introduced smooth probabilities. Durand[33], however, revised this approach for hidden Markov trees to avoid underflow when calculating $P(Z|X)$ probability matrices. To explain we need the following definitions:
$p(n)$ is the parent cell of cell $n$, and $c(n)$ is the children of cell $n$.
$\bar{X}$ is the observation of the whole tree and $\bar{X}_a$ is a subtree of $\bar{X}$ which is rooted at cell $a$.
$\bar{Z}$ is the complete hidden state tree.
$\bar{X}_{a/b}$ is the subtree rooted at $a$ except for the subtree rooted at cell $b$, if $\bar{X}_b$ is a subtree of $\bar{X}_a$.
For the state prediction we start by calculating the marginal state distribution (MSD) matrix. MSD is an $N \times K$ matrix that for each cell is marginalizing the

transition probability over all possible current states by traversing from root to leaf cells:

$$MSD(n,k) = P(z_n = k) = \sum_i P(z_n = k|z_{n-1} = i) \times P(z_{n-1} = i) \tag{4}$$

During upward recursion, the flow of upward probabilities is calculated from leaf cells to the root cells generation by generation. First, for leaf cells, the probabilities ($\beta$) are calculated by:

$$\beta_n(k) = P(z_n = k|X_n = x_n) = \frac{EL(n,k) \times MSD(n,k)}{NF_l(n)} \tag{5}$$

in which $X_n$ is the leaf cell's observation, and NF (Normalizing Factor) is an $N \times 1$ matrix that is the marginal observation distribution. Since $\sum_k \beta_n(k) = 1$, we find the NF for leaf cells using:

$$NF_l(n) = \sum_k EL(n,k) \times MSD(n,k) = P(X_n = x_n) \tag{6}$$

For non-leaf cells the values are given by:

$$\beta_n(k) = P(z_n = k|\bar{X}_n = \bar{x}_n) = \frac{EL(n,k) \times MSD(n,k) \times \prod_{v \in c(n)} \beta_{n,v}(k)}{NF_{nl}(n)} \tag{7}$$

where we calculate the non-leaf NF using:

$$NF_{nl}(n) = \sum_k \left[ EL(n,k) \times MSD(n,k) \prod_{v \in c(n)} \beta_{n,v}(k) \right] \tag{8}$$

and linking $\beta$ between parent-daughter cells is given by:

$$\beta_{p(n),n}(k) = P(\bar{X}_n = \bar{x}_n|z_{p(n)} = k) = \sum_j \frac{\beta_n(j) \times T_{k,j}}{MSD(n,j)} \tag{9}$$

By recursing from leaf to root cells, the $\beta$ and NF matrices are calculated as upward recursion. The NF matrix gives a convenient expression for the observation log-likelihoods. For each root cell we have:

$$P(\bar{X} = \bar{x}) = \prod_n \frac{P(\bar{X}_n = \bar{x}_n)}{\prod_{v \in c(n)} P(\bar{X}_v = \bar{x}_v)} = \sum_n NF(n) \quad n \in \{1, \dots, N\} \tag{10}$$

The overall model log-likelihood is given by the sum over root cells:

$$logP(\bar{X} = \bar{x}) = \sum_n \log NF(n) \tag{11}$$

Downward recursion: For computing downward recursion, we need the following definition for each root cells:

$$\gamma_1(k) = P(z_1 = k|\bar{X}_1 = \bar{x}_1) = \beta_1(k) \tag{12}$$

The other cells follow in an $N \times K$ matrix by writing the conditional probabilities as the summation over the joint probabilities of parent-daughter cell:

$$\gamma_n(k) = P(z_n = k|\bar{X}_1 = \bar{x}_1) = \frac{\beta_n(k)}{MSD(n,k)} \sum_i \frac{T_{i,k} \gamma_{p(n)}(i)}{\beta_{p(n),n}(i)} \tag{13}$$

Viterbi algorithm: Given a sequence of observations in a hidden Markov chain, the Viterbi algorithm is commonly used to find the most likely sequence of states. Equivalently, here it returns the most likely sequence of states of the cells in a lineage tree using upward and downward recursion[33].
The algorithm follows an upward recursion from leaf to root cells. We define $\delta$, an $N \times K$ matrix:

$$\delta_n(k) = \max_{\bar{z}_{c(n)}} \left\{ P(\bar{X}_n = \bar{x}_n, \bar{Z}_{c(n)} = \bar{z}_{c(n)}|z_n = k) \right\} \tag{14}$$

and the links between parent-daughter cells as:

$$\delta_{p(n),n}(k) = \max_{\bar{z}_n} \left\{ P(\bar{X}_n = \bar{x}_n, \bar{Z}_n = \bar{z}_n|z_{p(n)} = k) \right\} = \max_k \left\{ \delta_n(k) T_{k,k} \right\} \tag{15}$$

We initialize from the leaf cells as:

$$\delta_n(k) = P(X_n = x_n|z_n = k) = EL(n,k) \tag{16}$$

and for non-leaf cells use:

$$\delta_n(k) = \left[ \prod_{v \in c(n)} \delta_{n,v}(k) \right] \times EL(n,k) \tag{17}$$

The probability of the optimal state tree corresponding to the observations tree, assuming root cell is noted as cell 1, is then given by:

$$Z^* = \max_k \left\{ \delta_1(k) \pi_k \right\} \tag{18}$$

which arises from maximization over the conditional emission likelihood (EL) probabilities by factoring out the root cells as the outer maximizing step over all possible states.

*Fitting the cell phenotypes (maximization step)*. In the maximization step, we find the maximum likelihood of the hidden Markov model distribution parameters. We estimate the initial probabilities, the transition probability matrix, and the parameters of the observation distributions. The maximum likelihood estimation of the initial probabilities can be found from each state's representation among the root cells:

$$\pi_k^* = \gamma_1(k) \tag{19}$$

Similarly, the transition probability matrix is estimated by calculating the prevalence of each transition across the lineage trees:

$$T_{ij}^* = \frac{\sum_{n=1}^{N-1} \xi_n(i,j)}{\sum_{n=1}^{N-1} \gamma_n(i)} \tag{20}$$

where

$$\xi_n(i,j) = \left( \frac{\gamma_{p(n)}(i)}{\frac{\beta_n(i)}{MSD(n,i)} T(i,j)} \right)^T \times \frac{\beta_n(j)}{MSD(n,j)} \tag{21}$$

*Estimating emissions distribution parameters*. In the current study, we used two emissions distributions; first, a Bernoulli distribution for the probability of each cell fate, either at the end of each cell cycle phase or at the end of cell's lifetime; second, a gamma distribution for the durations of each cell cycle phase or overall cell lifetime. To estimate the distribution parameters after finding the cell state assignments, we calculated their maximum likelihood estimation weighted by their proportional assignment to that state. The initial and transition probabilities were shared across drug concentrations.

For estimating the Bernoulli distribution parameter for cell fate, we simply found the state assignment-weighted sample mean of the observations. To estimate the gamma distribution parameters, we fit all concentrations of each drug simultaneously and assumed that increasing drug concentration had a unidirectional effect on the observed phenotype within each state. This was implemented, using sequential least-squares programming (SLSQP)[67], through a linear constraint on the scaling parameter of the gamma distributions between concentrations so that higher concentrations had equal or greater average durations. The gamma distribution likelihood fitting is a convex optimization problem, indicating that local optimization can arrive at the globally optimal solution. Linear constraints do not change this property, and we confirmed fitting with different starting points arrived at the same solution. We used censored estimators to handle the effect of time censorship (explained below) in the duration distribution fitting. This was done by fitting uncensored and censored observations to the complete and survival distributions, respectively, and using the accumulated log-likelihood to estimate the distribution parameters.

*Baum–Welch*. Since both the hidden states and model parameters are unknown, we applied expectation-maximization (EM), known as the Baum-Welch algorithm in the case of HMMs, to find both the model parameters and cell states.

The expectation-maximization algorithm consists of two steps: expectation and maximization. During expectation, the probabilities of all cells being in specific states are calculated, such that for every cell and every state we have $P(z_n = k|X_n)$ and $P(z_n = k, z_{n+1} = l|X_n)$. The expectation step is calculated by the upward and downward recursion algorithms described above. In the maximization step, described above, the distribution parameters of each state, the initial ($\pi$) probabilities, and the transition probability ($T$) matrices are estimated, given the state assignments of each cell.

The expectation-maximization algorithm is initialized by randomly assigning the cells to states using a Dirichlet distribution. During fitting we iteratively switch between the expectation and maximization steps and then calculate the likelihood. If the likelihood improves less than a set threshold, we take that to indicate convergence.

**Model evaluation**. To find the most likely number of states corresponding to the observations, the Bayesian Information Criterion (BIC) was used[68]. The BIC requires the number of degrees of freedom, which we calculate using the number of independent parameters. Our model estimates a $k$ element initial probability vector, a $k \times k$ transition matrix, and a $k \times m$ matrix of state-wise parameters where $k$ is the number of states and $m$ is the number of parameters associated with observation distributions. For the phase-specific observation distributions we have a total of 6 parameters, including 2 Bernoulli parameters and 2 pairs of shape and scale parameters for the gamma distribution. Since the row-sums for transition and initial probability matrices must be 1, these values are not entirely independent. From distribution analysis of the phase lengths, we realized the shape parameter of the gamma distribution remains constant over different conditions, while the scale parameter changes. Therefore, the shape parameter was shared between the populations treated with 4 different concentrations of the same compound. Each condition, therefore, introduced 2 free parameters (1 Bernoulli parameter and 1 scale parameter). For the MCF10A experiments, terminal fates and cell cycle durations were also assumed to be Bernoulli- and gamma-distributed, respectively.

The shape of cell lifetime was similarly shared among the four conditions (PBS, EGF, HGF, and OSM).

The Wasserstein or Kantorovich–Rubinstein metric is a measure of distance between two distributions. This metric was used to determine the difference between state emissions[69]. An analytical solution, the absolute value of the difference in distribution means, was used for the gamma distribution.

**Model benchmarking**. We used emission distributions to represent the phenotypic characteristics of the cells within the lineages. To create our synthetic data, we considered two possible options as our set of observations throughout an experiment. In one case, we modeled the overall cell fate and cell lifetime; in the second, we modeled the phase-specific fate and duration. In both, we used a Bernoulli distribution for the fate outcomes and a gamma distribution for durations. The state assignment accuracy was calculated using the Rand Index[40]. The difference between true and estimated probability matrices was assessed using the Frobenius norm, or the sum of each element squared.

**Synthetic lineage data generation**. We generated synthetic lineage trees with $K$ discrete states and $N$ total number of cells for benchmarking. Lineages were composed of two primary data structures: the state and emissions trees. The state tree was randomly seeded with a root cell determined by the starting probabilities, then expanded by randomly sampling transitions based on the transition probability matrix. The lineages were extended by either increasing the number of initial cells, resulting in a greater number of lineages (breadth), or by lengthening the experiment time resulting in each lineage containing more cells (depth). After creating the tree of states with the desired number of cells, the emission tree is built upon it. Emissions were randomly sampled from the distributions for each cell's state. Finally, the effects of the emissions were applied to the tree when necessary. If any cells died, their progeny were marked as unobserved by making their emissions equal to NaN (Not a Number). If applicable, the effects of finite-duration experiments were also applied. Cells existing outside of the experiment duration were marked as unobserved, and those crossing the bounds of an experiment were marked as censored with duration clipped by the experiment.

*Time censorship*. Our phenotypic measurements include the cell fate (progression or cell death) and duration. These measurements are made for each cell cycle phase (G1 or S/G2) in the case of AU565 cells and for the entire lifetime for MCF10A cells. These measurements can contain incomplete information due to the bounds of an experiment. For instance, it is unknown when initial cells present at the start of the experiment began their cell cycle. The same is true of the cells present at the end of the experiment because we do not observe their end. Hence, a cell's lifetime and/or fate may be partially observed. To ensure our synthetic data is a close reflection of experimental data, we incorporated this effect in our synthetic data. Cells with lifetimes that extend beyond the end of the experiment were marked as censored for the lifetime estimation.

*Cell overall lifetime observations*. The parameters are reflective of the cell phenotypes we observed with 5 nM lapatinib treatment. Supplementary Figs. 1–5 are based on these parameters. Each figure includes 100 trials.

Transition probability matrix: $T = \begin{bmatrix} 0.9 & 0.1 \\ 0.1 & 0.9 \end{bmatrix}$

The initial probability vector is then calculated as the stationary distribution of states from transition probability matrix, satisfying $\pi = \pi * T$.

In this case, we have: $\pi = \begin{bmatrix} 0.5 \\ 0.5 \end{bmatrix}$

The same $T$ and $\pi$ were used for phase-specific emissions.

In Supplementary Table 1, "Bern p" refers to the Bernoulli parameter, the cell division probability at the end of its lifetime which is equal to 1—the probability of cell death at the end of its lifetime. "Shape" and "Scale" refer to the gamma distribution parameters. The cells' lifetimes were fit to gamma distributions.

*Cell cycle phase-specific observations*. The synthetic data used in Figs. 3, 4, Supplementary Figs. 8–10 were created based on the following parameters. These parameters are based on estimations from AU565 cells treated with 5 nM lapatinib. Each figure includes 100 trials.

In Supplementary Table 2, "G1 bern" and "S/G2 bern" are the cell division probabilities at the end of G1 and S/G2 phase, respectively. The "G1 shape" and "G1 scale" are the gamma distribution parameters of G1 phase lengths. "S/G2 shape" and "S/G2 scale" are the gamma distribution parameters of S/G2 phase lengths.

To benchmark the model with 5 states, we simulated 25–500 lineages, each with up to 31 cells, to create a population with 5 states. Like with the experimental data, we assumed the experiment ends after 96 hours and censored the cells' observations accordingly. The model parameters, including the transition probabilities and initial probabilities are listed below. The analysis results are

shown in Supplementary Fig. 14.

$$T = \begin{bmatrix} 0.6 & 0.1 & 0.1 & 0.1 & 0.1 \\ 0.05 & 0.8 & 0.05 & 0.05 & 0.05 \\ 0.01 & 0.1 & 0.7 & 0.09 & 0.1 \\ 0.1 & 0.1 & 0.05 & 0.7 & 0.05 \\ 0.1 & 0.1 & 0.05 & 0.05 & 0.7 \end{bmatrix} \quad (22)$$

$$\pi = \begin{bmatrix} 0.13 \\ 0.33 \\ 0.16 \\ 0.18 \\ 0.18 \end{bmatrix} \quad (23)$$

Figure 6 uses the first 4 states of Supplementary Table 3 as the parameter set for the emissions matrix to simulate varying state numbers in the BIC calculation.

*Varying emission differences.* To create synthetic data with subpopulations of varying dissimilarity (Fig. 5), we use the phase-specific parameters, with the values for the G1 phase gamma scale parameter for state 1 varying over [4, 20]. This results in an increase in the Wasserstein distance between the two cell states, allowing us to measure state assignment accuracy for different dissimilarity amounts between the two states. Likewise, for Supplementary Figs. 6, 7, we simulated the overall cell lifetime and varied the gamma distribution scale parameter from 1 to 8 for state 1.

*Optimal baseline classifier.* To compare the tHMM with a classifier that ignores heritability, we manually calculated the optimal classification boundary between the gamma distributions for state 1 and state 2. The best choice of classification boundary between two gamma distributions is the point at which the likelihood of the random variable, $x$, is equal between the two distributions:

$$p(x|G(k_1, \theta_1)) = p(x|G(k_2, \theta_2)) \quad (24)$$

where $k_1$, $\theta_1$, $k_2$, and $\theta_2$ are the shape and scale parameters of the gamma distribution corresponding to state 1 and 2, respectively. The shape parameter was shared between the two distributions. Consequently, this can be simplified to:

$$x = \frac{k \ln \frac{\theta_2}{\theta_1}}{\frac{1}{\theta_1} - \frac{1}{\theta_2}} \quad (25)$$

We assigned the classification labels to the observations using this classification boundary, which formed the baseline accuracy shown in Fig. 5c. As states 1 and 2 are identical at the very first point, we used the distribution mean ($k \times \theta$) as the threshold.

*Cross-validation.* To split the lineage data into train and test sets, we randomly selected 20% of cells from each condition and masked their observations such that they would not contribute to the fitting process. This was performed by setting the log-likelihood of the masked cells' observations to be uniformly zero for all the states. During the Baum-Welch fitting, the algorithm estimates the parameters using only the training cells. However, during the expectation step, the state of masked cells is still inferred via information about their relatives. After the fitting converges, we calculate the log-likelihood of the test cells' observations given their state assignments. This is accumulated into an overall likelihood of the held-out observations given the tHMM state assignments and fit.

To test this cross-validation scheme's ability to determine the optimum number of states for a cell population, we created synthetic populations with 2–5 true states. States 1–$n$ were used, where $n$ is the number of true states, to generate data that is like the experimental data. The state observation distributions shown in Supplementary Table 2. The transition probabilities were generated by adding 0.1 elementwise to the identity matrix and then normalizing it. The initial probabilities for all states were equal. Fitting was performed with models including 1–7 states. The optimum number of states was taken to be the smallest number of states at which the log-likelihood plateaus.

*Lowess trendline.* Locally Weighted Scatterplot Smoothing (Lowess) was used to provide the trendlines in the figures with repeated model runs.

**Statistics and reproducibility**. The experiments were repeated in three independent biological replicates and yielded similar results.

## Data availability
The experimental lineage data for AU565 and MCF10A cell lines can be found at https://github.com/meyer-lab/tHMM and https://doi.org/10.5281/zenodo.7195355 The synthetic data from which we plotted Figs. 3c, d, 4b–g, 5c–7 uses the code in the file named after the corresponding figure number. Data used in Figs. 7a, b, 8, 9 uses the AU565 cell line

experimental lineage data, and Fig. 7c and 10 use the MCF10A lineage data. The cell lines used in this study (AU565, MCF10A) can be made available upon request.

## Code availability
All analysis were implemented in Python v3.9 and can be found at https://github.com/meyer-lab/tHMM. The repository can also be found at Zenodo[70].

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

## Acknowledgements

This work was supported by the Jayne Koskinas Ted Giovanis Foundation for Health and Policy, NIH U01-CA215709 (A.S.M.), NIH U54-CA209988 (L.M.H.), NIH U54-HG008100 (L.M.H.). The authors thank Scott Taylor for his critical feedback that helped to improve the manuscript. The authors thank Ali Farhat, Adam Weiner, and Nikan Namiri for early exploratory work.

## Author contributions

A.S.M. and L.M.H. conceived of the study; A.S.M. conceived of the model; A.S.M, F.M., S.V. designed model; A.S.M., F.M., J.L., L.K., S.V. performed computational experiments; S.M.G. performed the experiments; F.M., J.L., L.K., S.M.G. conducted data analysis; A.S.M. and L.M.H. supervised the research; all authors wrote the paper.

## Competing interests

The authors declare no competing interests.
