## [Transparent Peer Review File · Communications Biology]

A lineage tree-based hidden Markov model quantifies cellular heterogeneity and plasticityResponse to Reviewers

We appreciate the reviewers' insightful comments about the impact of our work. We agree with the reviewers that this work provides a unique methodologic advancement when considering cellular plasticity in which the lineage relationships of cells is known. Our approach additionally enables the focused analysis of *phenotypic* heterogeneity, in contrast to defining cell states according to molecular features. We have endeavored to address each of the reviewers' questions, and are confident that they will find the manuscript substantially improved.

Reviewer #1 (Remarks to the Author: Overall significance):

The authors tailor the application of tree-based hidden Markov models to single-cell data where the lineage structure underlying the measurements is known. The authors show that this approach is useful to reconstruct cell-to-cell variability and dynamic changes in cell states. This work is for sure relevant in the context of methods for single-cell data analysis. The authors provide several, detailed benchmarks on synthetic data and analyzed experimental measurements in cancer cell populations. While for sure this paper provides a solid and conclusive story from the methodological point of view, I believe that the analysis of biological data could have [more extensively discussed] biological knowledge (or even more informed by biological knowledge), simply to understand if for example the inferred number of cell states is biologically sensible.

Reviewer #1 (Remarks to the Author: Impact):

I believe that this work will influence thinking in the field since it investigates important methodological achievements. Given that the main findings of the paper are methodological (although also well illustrated on some biological dataset), I believe that the most appropriate journal is Nature Communications.

Reviewer #1 (Remarks to the Author: Strength of the claims):

Here I provide some comments on how the claims could be better understood, supported, or discussed.

1. Discussion about the findings of the model in AU565 cancer cells. I see a discussion about the link between the phenotypic features uncovered and the known effects of Lapatinib and Gemcitabine, however I also think a discussion on the possible interpretation, and hence of the plausibility, of the 6 and 5 populations identified would be important. Can it be mapped into some existing cell classification? If not, what additional data should be gathered to have a more conclusive answer? Is there any similar, published dataset with a defined cell-state structure (derived, e.g., from transcriptomic profiling) where you could re-run your method and see if the inference of the number of cell states gives a sensible answer? Are the transition probabilities also justified from the biological knowledge?

We have considerably expanded our discussion of the resulting cell states, and their biological interpretation, within the discussion section. Briefly, we do think the phenotypic distinction between the cell states is biologically plausible, and that there is prior literature justifying the phenotypic and transition rate patterns we see with each drug treatment. As we elaborate on elsewhere in our response, we do not think there is an easy way to map these states onto existing cell classifications given the existing data here. Indeed, we think these cell states are likely reflect a confluence of both the drug treatment and cell-intrinsic properties and are therefore distinct from the clusters one would derive from just molecular snapshot experiments. Justification for this expectation is now mentioned in the introduction (lines 43–47). Most notably, a recent study marked cells based on their migration phenotype and discovered that they arrived at an entirely different clustering from what they would observe when just clustering molecular measurements [1]. Other studies looking at phenotypic

variation directly generally observe a small number of factors influencing phenotype, as opposed to the broad developmental programs that distinguish cells in omics profiling experiments [2]. As mentioned in the discussion, we plan to do experiments wherein we fix the cells at the end of their lineage tracking, and then assess different molecular markers in the exact same cells as were phenotypically profiled. We think this will help to resolve the underlying mechanism of these phenotypic differences. We think that this extends beyond the scope of the current analysis, however.

[1]: Chen, K. *et al.* Phenotypically supervised single-cell sequencing parses within-cell-type heterogeneity. *iScience* 24, 101991 (2021).

[2]: Spencer, *et al.* Non-genetic origins of cell-to-cell variability in TRAIL-induced apoptosis. *Nature* 459, 428–32 (2009); Kuchen, *et al.* Hidden long-range memories of growth and cycle speed correlate cell cycles in lineage trees. *eLife* 9, e51002 (2020); Mitchell, *et al.* Nongenetic origins of cell-to-cell variability in B lymphocyte proliferation. *PNAS* 115, E2888–E2897 (2018).

2. *I have several remarks on the figures.*

If the manuscript ends up in a more biological journal, like in Communications Biology, probably one should focus more the results in the main on actual applications to cancer data, I could suggest that figures 3, 4, 6 could be moved to the supplementary since they show benchmarks of the model's performance.

We very much appreciate this comment and agree that making this paper accessible to a non-modeling audience is important. However, we were conflicted in moving these figures to the supplement for several reasons. Firstly, they show some of the possible outputs from the model. Secondly, they demonstrate how one can verify the model will provide accurate results if applied to other forms of data. Finally, they provide a visual explanation for some of the key novelty within the manuscript. We have therefore elected to keep the figures within the main text. However, we have carefully gone through the text and removed technical language, replacing it with more accessible explanations. We have also added definitions where technical language is necessary. We hope this is helpful in making the paper accessible to a more biological audience.

c-d of Fig. 3 and b-c Fig 4: you could consider cropping the y-axis scale to a smaller range (like 0.5-1) to better show the variation, because it is concentrated on the upper part of the plot. In b-c Fig 4, it would clearer to have 'accuracy' as label of the y-axis. Please also add in the methods how such accuracy is determined. Why it seems that there are two solid lines like in blue and light blue (as well as green and light green)?

Thank you for these suggestions. We have cropped the y-axes indicating parameter values. We decided to keep the y-axes indicating accuracy from 0–100% to avoid any of the potential confusion cropped axes can create. We renamed the y-axes to “Random Index Accuracy” to reflect how they are calculated. We also added an explanation of how the accuracy is calculated to the figure caption and methods. The light green and blue solid lines show the true value of the parameters and are now explained in the caption.

Where is in b-g the variability in depth and breadth showed in 4a? Same question for Figure S1, S2 and S3.

Apologies this was not clear. Extending the “depth” of the dataset corresponds to running the experiment longer so that we have a greater number of cells within each lineage tree. Extending the “breadth” instead corresponds to tracking a greater number of lineages. We have now added arrows to indicate this in the figures, defined these concepts in the captions (lines 208–210), and explained the different trade-offs in data generation in the methods (lines 595–597).

- Fig. 3-5: add in the caption what dataset is used, i.e., like '2-state synthetic dataset'. Please also add how many repetitions of the inference are plotted (e.g., in Fig 5c).

An indication of the dataset and number of repetitions has been added to the captions for figures 3-6 and line 643–664 in the methods where we explain figures S1-S10.

Fig. 5c: how are the bunch of blue points around accuracy 40% explainable? Does k-means also have a lower performance here? It would be probably interesting to understand to what extent lower-performance repetitions for tHMM are also low-performance for k-means hence whether they are due to the ambiguity of that particular tree realization. Please also explain what the Lowess trendline is.

Thank you for highlighting this strange behavior. We also found this difficult to explain, because the benefit of the tHMM model's tree formulation should be consistently present. We traced these poor results to an interaction between the way we matched the tHMM states to the underlying "true" ones, particularly when the states are misbalanced in their abundance. We corrected this issue by using to the random index accuracy which does not require that we match the tHMM and "true" state identities and is more appropriate for quantifying the accuracy of clustering results such as these.

We have added an explanation of for the Lowess trendlines (lines 677–679).

Fig. 6: what are the 'histograms' here? Do you mean the different lines? I am afraid that the word histogram is misleading here.

Indeed, this was unclear. We produced a histogram of the model-selected number of states on the figure as a light shading behind the AUC value lines. However, because the model almost always picked the right number of states, it was not obvious this was a histogram. We removed the histograms from Figure 6 to avoid any confusion.

Fig 7a: how can you clearly distinguish 6 states from 7 states? Shouldn't I see 4 curves (one for the control and one for each concentration used)? The latter remark applied also to Fig 7b.

Thank you for pointing out this unclear point. Indeed, based on the BIC metric both 6 and 7 states are of equal quality in fitting the lapatinib data. However, we have selected the 6-state model as it is most parsimonious. We have added a note about this decision on lines 276–279.

We assume that each cell state is preserved across concentrations of each drug. Both the initial abundance of each cell state and their transition rates are therefore shared across all concentrations of a drug. Because we fit all concentrations of a drug as a single model, we perform the model selection procedure for a single model across all concentrations of a drug. Therefore, there is one line on the plots in Fig. 7. We have added an explanation of this point to the result's text that explains figure 7.

3. I have also a few doubts concerning the methods' choice and explanation.

One of the main features of the pipeline you describe is the use of a 'censored' model. Please state what 'censored' means and, in the Methods, how it is achieved.

We have added an expanded explanation of censoring on first use of the term (lines 153–163) and a dedicated section in the methods (lines 604).

It is probably worth having numbers for your equations.

We have added these.

In the sentence: 'The E-step is calculated by the upward and downward recursion algorithms.' I would add '(see below)', otherwise it is not clear that these upward and downward recursion algorithms will be discussed thoroughly.

We have added this (line 478).

It seems that you use $\beta(n,k)$ and $\beta_n(k)$ to denote the same quantity, please clarify or make the notation consistent.

Thank you for identifying this typo. It has been fixed (line 505).

In the expression for $\delta(n, k)$ it is not clear where the dependence on k stems from (I see rather a j that could be a k ?)

We have fixed this typo as well (line 550).

How are the choices of the Gamma and Bernoulli distributions justified?

Thank you for pointing out this oversight. For the cell cycle durations, we evaluated a variety of distributions and found the data matched a Gamma distribution best. We were also inspired by a previous paper that very thoroughly justifies use of a Gamma distribution as a representation of cell cycle durations [1]. We have amended our explanation of this point in the manuscript (lines 194–196).

The Bernoulli distribution arises any time one has a binary process of defined probability—it is difficult for us to determine what would be an alternative here. We do assume independence between the cell cycle phase durations and outcomes since we handle these two distributions separately. We note this point in the manuscript (lines 579-585). We have not observed any noticeable correlation between these two quantities, and accounting for both distributions jointly would be extremely complex, so we think this assumption is reasonable.

[1]: Chao, H. X. et al. Evidence that the human cell cycle is a series of uncoupled, memoryless phases. *Mol Syst Biol* 15, e8604 (2019).

Reviewer #1 (Remarks to the Author: Reproducibility):

The tests for validating the approach are appropriate and, apart from a few doubts I indicated in the comments, the methods are described in detail enough to be reproduced. The code also is made available.

Reviewer #2 (Remarks to the Author: Overall significance):

In this manuscript, Mohammadi et al utilize a tree-based hidden Markov model to learn patterns of single cell heterogeneity and cell state transitions based on single cell lineages. They use this approach to capture the evolutionary dynamics of drug response in tumor cells treated with chemotherapies (using synthetic data or experiments in a breast cancer cell line), infer the number of phenotypically distinct subpopulations, and classify cells based on their phenotypic heterogeneity using measurements of cell cycle progression. The hidden Markov models used in this study provide an efficient tool to infer discrete states from measurements when a series of co-dependent observations are made. This provides an advantage over population-based analysis of drug response or fixed-time single-cell assays that generally miss the contribution of individual cells, thereby allowing the study of intrinsic or drug-induced plasticity in heterogeneous cell populations.

The overall topic of the manuscript is interesting. The single-cell measurement and analysis methods utilized in this study to investigate the problem of cell heterogeneity and plasticity are informative. The diversity of quantitative outcomes inferred from this approach makes it different from previously

described analysis tools. These include the inference of starting and transition probabilities of cell states, the distribution of cell phenotypes (e.g., their growth rate) in each state, and the likelihood of observations, which can be used to estimate the number of distinguishable cell states. It is noteworthy that performing unbiased time-lapse experiments on populations of heterogeneous cells is not trivial, and the analysis of incompletely observed cells typically imposes a systematic bias in data analysis. However, the authors have noticed this issue and (instead of removing such cells) have used a censored estimator and the survival function of the distribution (calibrated against synthetic data) to correct their estimations of the cell lifetime properties and survival versus death for these cells.

Reviewer #2 (Remarks to the Author: Strength of the claims):

Despite the methodological strengths mentioned above, I have some questions about the manuscript findings and its biological consequences that I think the authors should either clarify or address to further strengthen the case for publication:

1) The authors explained that their analysis cannot identify the molecular factors leading to these cell states. They suggest that single cell molecular analyses such as transcriptional profiling, tied with molecular barcodes, should allow them to align molecular and phenotypic states in the same populations of cells. However, those methods would probably be able to identify diverse cell states without performing laborious time-lapse experiments and single-cell lineage tracking by imaging. I think the major benefit of the methods described in this study would be to extract other dynamic live events (using, for example, multiplexed reporters of cell signaling) associated with the emergence of cell state transitions and drug response heterogeneity. I would enjoy a deeper discussion of the potential applications of their method for future studies by the authors and other researchers who are interested in the field of tumor heterogeneity.

Indeed, we had a notion that one might be able to align molecular features using a combination of single cell sequencing and molecular barcodes. To investigate this, we performed a computational experiment wherein we tracked how barcodes would overlap with cell state over time. This helped us to recognize that, in fact, barcodes would provide essentially no information about how single cell measurements of the population align to cell state. This is because barcoding experiments generally rely on a bottleneck in which there is selection of a small set among the full barcode diversity. Because cell states are dynamic, this bottleneck does not occur, and barcodes of a single state eventually mix to be represented among all states.

An alternative strategy is experimentally more challenging but guaranteed to provide useful information. What we can do instead is image cells to collect their phenotypic responses, and then measure the molecular features of those exact same cells at the end of the experiment [1]. This way, we know the state assignment of each cell and therefore can evaluate the average expression or abundance of molecules in each state.

We have amended our discussion section to reflect this new perspective.

[1]: Lane K *et al.* Measuring Signaling and RNA-Seq in the Same Cell Links Gene Expression to Dynamic Patterns of NF- κ B Activation. *Cell Syst.* 2017 Apr 26;4(4):458-469.e5.

2) By Integrating the experimental and analytical methods, the authors were able to identify 6 and 5 distinct subpopulations within the lapatinib and gemcitabine-treated data within 96 hours. This timeframe covers approximately 5 generations of cells. I understand that individual live cell tracking beyond 96 hours is challenging. However, I wonder about the number of cell states that could be expected by extending the length of these experiments? Do we expect to see a plateau or continuous increase in the number of cell states with time? Acquisition of additional genetic mutations is expected in cancer cells, but not all genetic mutations are phenotypically consequential.

Great question! Each state we identify is primarily defined by having a unique phenotype. Therefore, we would expect the number of cell states to quickly plateau and, if we ran the experiment for longer, we expect we would likely end up with the same number of states or perhaps one more. It is true that if you ran the experiment for much, much longer, you could end up with additional states that are primarily defined by transition dynamics differences, such as from mutational processes. However, just like you point out, most mutations are not going to have phenotypic effects. Because our states are defined by phenotype, anything that does not affect the phenotype of the cells will not affect the number of states. A corollary of this is that the number and type of states identified is dependent on the phenotype being analyzed. We think this is a potential strength as it allows one to focus on specifically whatever variation is phenotypically consequential.

3) The authors explained that the model is agnostic as to whether the heterogeneity is pre-existing, or drug induced. Is it possible to determine or infer the factors that affect the number of these states? Besides time (discussed above), is it possible to comment on whether these states are unique to cancer cell populations (the AU565 cell line in this case), or are these cell states determined solely based on the mechanism of action of drug?

States represent distinct phenotypic responses within the cell population. Therefore, we expect that the number of states reflects the interaction between cellular factors and drug treatment. That said, we expect that states will be largely overlapping between different cell lines for the same drug, because the differences in drug treatment response are probably a function of a small number of factors determined by the drug mechanism.

One thing we can say at this point is that this heterogeneity is not limited to cancer cell populations. We have expanded our analysis to include the non-tumorigenic cell line MCF10A and again observe heritable variation in proliferation and growth factor responses, albeit with fewer inferred states (lines 325–334). While cancer cells have additional genetic variation, we think that much of what we observe is not determined by mutational changes.

For future work, we are now expanding our data collection to several cell lines and a wider panel of drugs to answer these exact questions more definitively.

For example, if similar experiments are performed on a mixture of cells from two distinct cell lines that each reveal 6 states with lapatinib treatment individually, do we expect to find 6 states, 12 states, or a number between 6 and 12?

We expect that in this situation you would identify 6–8 states from the mixture. It is true that two cell lines can respond very differently to drug treatment. However, the number of states is bounded by the number of distinct drug responses one can identify. We would expect that two cell lines are largely overlapping in their states and any additional states would mostly help to account for differences in state abundances and transition dynamics. We do plan to collect measurements from additional cell lines very soon to investigate this point.

Reviewer #3 (Remarks to the Author: Overall significance):

The current study establishes a tree-based adaptation of a hidden Markov model (tHMM) that allows lineage tracing single-cell dividing dynamics to demonstrate heterogeneity in tumors and drug response. While it is an interesting [computational] model, [the] biological significance [of the] different cell states is not clear. There are several major flaws with the [MS].

1) The authors referred to (in their reference No.57) and applied a cellular model which was previously reported in “Single cell tracing reveals heterogeneous drug-, dose-, and time-dependent effects on cancer cell fates” (Sean M. Gross et al., Cold Spring Harbor Laboratory, 2020.) to track cell cycle

lengths in cell lineages with or without gemcitabine. And then used a lineage tree-based hidden Markov model (tHMM) to fit these measurements.

The authors claimed that “This work provides a flexible phenotype driven route to discovering cell-to-cell variation in drug response, demonstrates an overall strategy for quantifying the dynamics of cell heterogeneity, and implements a very general software tool for the widespread use of tHMM models.”

However, for the first two points:

1. route to discovering cell-to-cell variation in drug response,

Apologies, but we are unable to understand the reviewer’s point here. We would appreciate any clarification.

2. strategy for quantifying the dynamics of cell heterogeneity were already reported by Sean M. Gross et al. [In fact] Figures 1 and 2 from the current paper are almost identical to figure 3 from Sean Gross paper.

Indeed, the underlying imaging data used in this paper was collected during experiments described in Gross et al. However, our analysis here is entirely based on single cell tracking, while that of Gross et al is based on population-level observations. The approach, model structure, goals, and findings are completely different.

Apologies, but we are unable to identify any similarities between Fig. 2 of this paper and Fig. 3 of Gross et al. Figure 1 in the current paper has some superficial similarities to Fig. 3A from Gross et al, but this figure is simply used to illustrate how the data was collected. Analogously, one would not be surprised to also find similar pictures of a microscope in both studies, but that does not mean that all microscopy studies are redundant.

2) The experimental design lacks essential controls such as cell cycle synchronizing.

Experimental controls serve to ensure the conclusions from an experiment are not an artifact arising from unintended sources. Based on this definition, we do not believe that synchronizing our cells would be a useful control. Indeed, we believe it would likely introduce artifacts in our analysis.

Our model does not require or assume cells are synchronized at the beginning of the experiment. As we describe in Fig. 3, we modify the tHMM fitting process to recognize cell cycle periods whose lifetime is influenced by the boundaries of the experiment. These are handled with the censorship considered.

In contrast, cell synchronization is a disruptive procedure that has well-documented unintended consequences on cell behavior [1–3]. We are not confident that the behavior of cells after synchronization would be reflective of their true behavior upon drug treatment.

[1]: Gong et al., "Growth imbalance and altered expression of cyclins B1, A, E, and D3 in MOLT-4 cells synchronized in the cell cycle by inhibitors of DNA replication." *Cell Growth and Differentiation*. 1995 Nov;6(11):1485-93.

[2]: Darzynkiewicz et al., "Cell synchronization by inhibitors of DNA replication induces replication stress and DNA damage response: analysis by flow cytometry." *Methods Mol Biol*. 2011;761:85-96. PMID: 21755443; PMCID: PMC3137244.

[3]: Grolmusz et al., "Fluorescence activated cell sorting followed by small RNA sequencing reveals stable microRNA expression during cell cycle progression." *BMC Genomics*. 2016 May 27;17:412. PMID: 27234232; PMCID: PMC4884355.

The results also lack quantifications and statistical analysis, making it difficult to appreciate the findings.

Apologies, but we are unsure exactly what form of additional statistical analysis the reviewer might expect. We are proposing a new statistical approach to analyzing this form of data, and so have concentrated on validating the accuracy of the model. The complex form of these data precludes the use of simple statistical tools. For example, one cannot use a simple univariate null hypothesis test to determine whether two states are significantly different, because the state properties are dependent on one another. Therefore, we use a model selection strategy as described in Figure 6 and 7.

In this paper, the biological aspect of the model is only based on one cell line, the AU565, and a short-term tracing, 96 hours, which is not strong enough to state a conclusion of demonstrating heterogeneity and lineage histories in tumors. Additionally, the current study oversimplifies the complexity of the heterogeneous tumor system, which was only reflected in diverse cell dividing time courses and several unidentified cell states.

Indeed, we do not claim we are modeling the full history or complexity of the *in vivo* tumor environment. That said, simplified model systems are essential for studying the mechanistic processes within these much more complex systems. We maintain that the approach here is a useful tool for studying how there can be such a remarkable amount of heterogeneity even within an extremely simplified system like this one.

3) Additionally, the authors used Incucyte to: (1) [Collect] signals from [the] cell cycle reporter. (2) [M]anually track cell lineages. This may be problematic as Incucyte may automatically adjust fluorescent signal strength in images based on signal/background ratio, and the cells may move into/out of image fields. Therefore, other methods should be applied to validate the quality of data before a model such as tHMM is to fit these measurements.

Thank you for highlighting that we were unclear in how the cell cycle reporter works. This reporter is measured based on whether there is greater fluorescence within the nucleus or cytoplasm [1]. As a result, the reporter signal is invariant to changes in exposure and background. We have made a note of this point on lines 427-429.

We agree that cells entering or leaving the field of view is a potential source of bias. However, like many HER2+ cell lines, AU565 cells are not motile, and essentially no cells move more than a cell length across all our images. To investigate whether this might still be a concern, we quantified the number of cells that are close enough to potentially leave the field of view during an experiment due to their proximity to the image boundary. This was less than 4% of the total cells across an image. These observations ensure that there is no sampling bias due to the limited field of view. We have noted these observations on lines 443-445.

[1]: Spencer *et al.* The proliferation-quiescence decision is controlled by a bifurcation in CDK2 activity at mitotic exit. *Cell*. 2013 Oct 10;155(2):369-83.

In general, substantial work is for this work [...] to be published, which may include but not limited to:

1. At least to establish another cellular model to strengthen novelty and generality.

Thank you for this suggestion. We have added a second cellular model in a different treatment context, MCF10A cells treated with a panel of growth factors. We chose this second model to also help address some of the questions from reviewers 1 and 2 about the sources of state differences, since these cells are non-tumorigenic. These results can be found in Figures S13 & S14 and are discussed in the last subsection of the results (lines 325–334).

2. Need to sufficiently characterize the [...] cellular models, particularly if the image-based cell cycle tracking system under current experimental settings can faithfully represent cell cycle distribution measured by [a gold] standard [method] (such as by DNA content and DNA synthesis.)

This method of tracking the cell cycle has been extensively characterized in prior work [1]. Apologies that this was not clear in our original submission. We have amended our description of the reporter system (lines 423-429).

As additional validation of the reporter, we fixed and stained cells from one of the conditions reported in the manuscript. This allowed us to compare the fraction of cells reported in each cell cycle phase using the reporter or a more typical analysis of DNA content. In both cases, we see a bimodal distribution with essentially identical numbers of cells assigned to each phase.

[1]: Spencer *et al.* The proliferation-quiescence decision is controlled by a bifurcation in CDK2 activity at mitotic exit. *Cell*. 2013 Oct 10;155(2):369-83.

3. Need to validate lineage tracking by a more rigorous method, such as lentivirus barcode library followed by single cell sequencing.

Cell tracking through microscopy images is widely accepted as a rigorous means to trace cell lineages [1]. As we now discuss more verbosely in the manuscript's discussion section and above (second reviewer's first question), a barcoding library would not provide similar information.

[1]: Hilsenbeck, Oliver, et al. "Software tools for single-cell tracking and quantification of cellular and molecular properties." *Nature biotechnology* 34.7 (2016): 703-706.

Specific critiques:

In Figures 1a and b, two replicate controls of untreated AU565 cells demonstrate very different dividing dynamics without interpretation.

Apologies that this was not clear in the figure caption. Figures 1a and 1b are not different replicates; they are in fact lineages pulled from the exact same experimental condition and replicate! The differences are purely from sampling a different subset of the data. We too were struck by the heterogeneity found upon this exercise. To clarify this point, we have added a note to the caption that these lineages come from the same replicate.

The cell cycle phases durations, doubling times, and other cell division parameters need to be quantified and statistically compared between each condition (also in Figure 8 and 9, S11 and S12).

Apologies, but we are unsure what would be the use of these average quantities *between* conditions, since all comparisons within the paper are between populations of cells *within* conditions. We would appreciate some clarification if we have missed a use for these quantities.

In addition, all the cells are required to be initially synchronized.

Please see our discussion of cell synchronization above.

In Figures 8 and 9, different states of cells need to be identified and biologically characterized, and the state transition intra- and inter-lineages mechanisms are unclear.

Our goal in this work was to develop a model that can accurately identify the phenotypic heterogeneity within and between cell lineages. To reiterate, we present (1) a unique experimental dataset in which the phenotype and relationships of cells are tracked during perturbation and (2) a new computational approach to identifying heritable variation in these populations. Future work will identify the underlying biological mechanisms of these states and their transitions.

Each biological replicate needs to be labeled in all lineage trees to exclude replicate-dependent cell [state differences].

Thank you for this important suggestion. The results shown in the manuscript are from the union of three replicates. To determine whether cell states might reflect replicates rather than biologically meaningful differences, we separated each replicate after fitting and plotted the abundance of each state in each replicate for all conditions, as shown below. We see essentially no relationship between replicate and cell states. We have added a note to the methods about this procedure (lines 445-447).

Additional cell lines/tumor tissues and longer time tracing are suggested to validate the tHMM model further.

While we appreciate the suggestion, we maintain that the present work is an important and substantial advance. We certainly plan to expand the samples and timescales investigated in future work.

We thank the reviewers for this second round of feedback that has again strengthened the conclusions and description of our study. We wish to highlight that all three reviewers find our approach to be an elegant solution to an important problem of general interest. Please find a point-by-point response to each comment included below.

Reviewer #2 (Remarks to the Author: Overall significance):

The authors have made significant improvements in the description of their experiments and data presentation [and], in my opinion, have addressed the reviewers' comments satisfactorily.

We thank the reviewer for their helpful comments in improving our work.

Reviewer #4 (Remarks to the Author: Overall significance):

The manuscript describes an interesting and important concept, the inclusion of cell history in the definition of cell states, in the analysis of cell line heterogeneity and single cell states. Overall, the description of the work is concise and clear, although probably not very easy to follow for cancer biologists. Although the general message is of general interest, the complexity of the description of the method and analyses seems more appropriate for a specialized journal.

We appreciate the reviewer's enthusiasm for the general importance of the problem addressed in our work. Indeed, the requisite analysis is quite complex. However, we have tried to demonstrate the value of the method through its application. Also, as described in the manuscript, we have taken several steps to make this analysis accessible to the wider computational systems biology community.

Reviewer #4 (Remarks to the Author: Impact):

The work would gain in significance if (an)other cell line(s) would have been included with the same treatment. This would allow for broader conclusions regarding cell line intrinsic characteristics and treatment specific effects or "unique cell states". The authors agree on the importance for the inclusion of other cell lines, in the rebuttal, and this seems crucial in the interpretation of numbers of unique states. [Considering] this, it would also be informative to use treatments with drugs that at low dose cause a cell cycle arrest and at high dose cell death. This would provide a clearer picture of cell states leading up to these very different responses, including the pre-treatment history. The extension with the analysis of MCF10A cells under growth factor deprived conditions seems less relevant for the broader implications of this work.

We agree these are exciting future directions and look forward to implementing them in future studies. However, we agree with Dr. Inglis that this is outside the scope of the present manuscript.

Reviewer #4 (Remarks to the Author: Strength of the claims):

The authors indicate that the inclusion of other types of molecular data, including mRNA expression or other reporters, would strengthen the observations and conclusions described in the manuscript. Although one can argue that this is beyond the scope of this work, the techniques required to perform such analyses are widely available and feasible.

Indeed, we agree that the techniques for incorporating molecular data are feasible. As described in our response to the first round of reviews, we do plan to perform end-point molecular measurements tied to lineage tracking to expand upon this work. However, we maintain that the work presented here already provides a novel analysis into the phenotypic heterogeneity of cell populations. Our planned end-point experiments, while feasible, are a challenging undertaking that requires substantial assay development work. Including detailed molecular analysis would result in an unwieldy study and detract from the existing findings.

Minor comment:

Figure 2C; “state 1” rather than “1 state”.

We have fixed this notation.

Reviewer #5 (Remarks to the Author: Overall significance):

The authors present a method to take advantage of single-cell level lineage tracing to identify stereotypical phenotypic states that the cells transition between. This method, which employs a tree-based hidden Markov model, provides an elegant means to dissect the evolution of heterogeneity. The authors provide a compelling range of benchmarks on synthetic data. However, what is less convincing is their leap from results on pristine synthetic data with two states, to real world cellular data containing cells in a range of drug concentrations, where their assumptions may not be met, in which they detect 5 to 8 populations. Additionally, given the nature of the features used it is (understandably) hard to interpret the significance of the cell-states detected. Thus, I think the work would be significantly strengthened by solidifying the results on real data, either via new experiments with better controls, synthetic simulation that are closer to the real-world data and an additional discussion of the assumptions/limitations.

Thank you for these suggestions. These points are reiterated later and so we have included a more detailed response below.

Reviewer #5 (Remarks to the Author: Impact):

As a method and idea, I feel this paper has the potential to be very impactful. Especially in the context of image-based assays, where lineage tracking is possible, I think this idea could really catch on. Future iterations where a multivariate version allows tracking of morphologic and signaling phenotypes could reveal very interesting insights. The biological results primarily serve as a proof of principle, although there may be some interesting hypotheses to be followed up in future experiments.

We greatly appreciate the reviewer's enthusiasm for the impact of our approach. Indeed, we are following up on our findings here in several different ways. First, we are setting up an automated pipeline to be able to quantify these other aspects such as morphology and migration alongside lineage history. Second, we are applying endpoint Nanostring measurements to obtain molecular information about the cells at the end of the experiment, so that we can identify molecular features of each state. Both studies are made possible by the analysis developed here.

Reviewer #5 (Remarks to the Author: Strength of the claims):

The claims on synthetic data are mostly sound. However, I took issue with the assertion that superior state prediction of their method compared to k-means "shows that heritability can help to identify cell states more accurately with partially overlapping phenotypes.". K-means is fine as a baseline for what a naive approach might be. However, even without the heritability information it makes use of a far less sophisticated analysis: e.g., no fitting of features to individually curated distributions etc. Thus, the improvement could be attributed to these differences rather than the inclusion of lineage tracking. An apples-to-apples comparison would use a similar clustering methodology, just without the lineage information. Additionally, I could not find a description of how k-means was performed. The variables they use exist on completely different scales, and thus appropriate normalization must be used before applying k-means.

Thank you for pointing this out. We agree that there was the potential for reduced performance from k-means due to mismatch the the assumptions of the algorithm. To resolve this concern, we replaced k-means with an analytical approach in which we determined the optimal discriminatory boundary using the true distributions. As this is the *optimal* classification possible without using the cell-to-cell relationship information, we feel this provides a much more convincing comparison baseline. We explain this analytical approach in detail in the methods (lines 746 – 759) and have updated figure 5 accordingly.

My primary [concern is] that the biological data sets stray quite far from what was shown with the synthetic data.

Almost all demonstrations with the synthetic data used 2 cell states, whereas the biological data discovers on the order of 5 to 8. How reliable are the identifications of individual states in this scenario given the number of [cells] available? This could be clarified with further simulation.

This is a great suggestion. We have added this benchmarking using a synthetic population of 5 states using parameters that are reflective of what we see in the experimental data (Fig. S15). While classification with more groups can become challenging because there are more opportunities for the wrong assignment, we see comparable performance with 5 states as compared to 2. While not included in the manuscript, we also examined the confusion matrix of state assignment outcomes and see, as expected, that the small fraction of incorrect assignments preferentially occurs by confusion between the most similar states. This suggests that, even when the model leads to incorrect assignments, it is simply due to overlap between the distribution defining the two states, exactly as would occur with any clustering strategy.

How are we sure this approach works on real world data? Does the Markov assumption even hold? I could imagine a scenario where multiple divisions give rise to local clumps of cells, thereby altering their nature/transitions relative to the parents. This seems like something that is testable. Would be good to build up confidence through such demonstrations.

This is a good point that we could have addressed more directly in the manuscript. There is both a biological and analytical argument to be made for the relevance of the Markov assumption with this problem.

Assuming a Markov process comes down to the assumption or simplification that the behavior of a cell can be conditioned only on the state of its parent. In other words, cells are conditionally independent of their grandparents, should we know the state of the parent. Note that very complex behavior can still be expressed within the framework of a Markov assumption. For instance, the example given, with a clump of cells, would still be reasonably represented because the parent carries with it the clump status, and therefore there is no information lost due to conditionally disregarding the grandparent cell.

In a survey of related studies wherein non-genetic inheritance has been explored, we have not been able to identify inheritance patterns that would be inappropriately accounted for by the Markov assumption [1, 2]. Kochen *et al* is an illustrative example [2]—while there are multi-generational interactions, these can be accounted for through cycles of cell states, exactly as we observe. For the Markov assumption to be violated, there would have to be an effect of previous generation cells *that cannot be accounted for through the state of the parent cell*. We have trouble even imagining what this might be.

We think one reason that the Markov assumption may not be as readily accepted here is its unfamiliarity relative to other modeling techniques. It is important to note that *any* system that can be represented as an ordinary differential equation can be modeled through an equivalent Markov process. Saying that the Markov assumption is inappropriate is equivalent to saying that an ODE model would be inappropriate, and one needs to explore an alternative modeling scheme (e.g., delay differential equations). The universal utility of ODE models should be indicative of how widespread the utility of assuming a Markov process is.

Indeed, in other areas such as time-series analysis, there are not well-developed schemes for even testing whether a Markov assumption is appropriate. Relaxing this assumption quickly leads to models that are too complex for parameterization, and so this assumption is used as a fundamental tenet of any modeling strategy. While we cannot come up with a scheme for testing this assumption, beyond implementing a second-order model which would be infeasible, we think the most convincing argument for this assumption is the inability to even imagine biologically plausible schemes that would violate this assumption.

While we do not demonstrate a direct test of the Markov assumption, we designed cross-validation scheme to demonstrate that the model is sufficiently accurate to predict held-out data. This might be the most rigorous validation of the Markov assumption that we can offer. Briefly, we masked ~20% of the cell observations, and then used their relationship to non-masked cells to

predict their behavior. This scheme showed that inheritance is necessary and sufficient to predict the behavior of *unobserved* cells (Fig. S16).

We have added some of this argument to lines 69–75, 292–305, and 759–776.

[1] Cassidy *et al.* "The role of memory in non-genetic inheritance and its impact on cancer treatment resistance." *PLoS Computational Biology* 17, no. 8 (2021): e1009348.

[2] Kuchen *et al.* "Hidden long-range memories of growth and cycle speed correlate cell cycles in lineage trees." *Elife* 9 (2020): e51002.

More broadly, I think the paper would benefit from some real world "Controls": clear cut ground truths on straightforward data that the method should recover. Examples might include mixing two cell lines and ensuring there are no transitions between these (although in their response to Reviewer #2, the authors seem to rule out this). Alternately, are there additional excluded variables between which we should never see transitions (e.g., morphological states?). Admittedly some of these are out of the scope of the present work.

We completely understand the desire to have some sort of experimental controls that somehow lack the full complexity of the real experimental samples. However, after carefully considering options for something like this over several years, we have concluded that there are really no opportunities to generate something like this for this type of data.

What the reviewer desires, if we understand the comment, are some collection of cell measurements where one would not expect to observe heritable heterogeneity. An analogous example in population-level measurements would be a sham treatment for which one does not expect to observe a cell response. However, in contrast to this example, the responses measured here arise both from external perturbation and natural variation within the cell population.

We were tempted by the suggested cell mixing experiment previously. However, like we described in the response to reviewer #2, our data directly measures the cell relationships to one another. Therefore, our model's inferences do not relate to the relationships between cells—that is the data itself. This contrasts with snapshot single cell experiments, where the similarities between cells are indeed inferred (e.g., pseudotime methods), and so a mixing experiment would be more informative.

We also do not think it would be useful to search for some measurements where cell heterogeneity or transitions do not exist. It is essentially impossible to demonstrate there is no heritable heterogeneity in a population of cells and, indeed, is unlikely to be the case given the molecular drift inherent in living systems. A "control" lacking transitions would be equivalent to the cell mixing experiment above—the experimental data would be trivial since we explicitly track cell relationships, and so it would not benchmark the model in any way.

While we cannot remove the variation arising from natural variation within the cell population, we *do* show that experimental replicates arrive at a similar mixture of cell states (shown in the first

reviewer response, described in lines 495–497). This control does allow us to conclude that, upon repeating the experiment, we arrive at similar conclusions. Additionally, as described above, we have added a much more rigorous cross-validation strategy to demonstrate the model is a sufficiently accurate representation of the data to predict held-out data.

Ultimately, the only route to more convincing evidence, beyond the various statistical justifications that we have provided, would be to identify the molecular mechanism underlying our observed phenotypic heterogeneity. We think, however, that the reviewer understands the challenge in doing this, and that it lies outside the scope of this present manuscript.

We have added some of these points in the discussion (lines 408–427).

I am particularly concerned by the leap in complexity, and addition of several little discussed assumptions, in combining multiple concentrations of drugs into a single analysis. This is important, as deciding whether it is the same states whose properties slowly change with concentration is a subtle business. See following claims:

"The model was then fit to each experiment's data across all conditions, enforcing that the initial and transition probabilities are shared across concentrations while allowing the phenotype distributions to vary." It is not clear to me that increasing the amount of drug would not lead to a change in transition between the different states.

Thank you for pointing this out. This is certainly a possibility, and one that we considered. We ultimately settled on creating shared transitions and initial probabilities matrices because this provides a basis by which we can identify correspondence in the cell states between conditions. Allowing these quantities to vary between conditions would mean that there is essentially nothing shared between drug concentrations, and so "state 1" in the first concentration could be any of the states in the second concentration, and an alignment would have to be performed.

As a second practical matter, there are very few division events observed in the higher drug concentrations, and so those conditions have few observations on which to define the transitions matrix; conversely, the transitions matrix has little effect on fitting that data. This does mean that the transitions matrix is mostly a reflection of the lower drug concentrations and control condition. Collecting data over longer experiments would expand the number of division and therefore transition events observed, but there are practical limitations to setting up week or longer imaging experiments.

However, we agree that it is important to show this decision does not compromise the accuracy of the model. To do so, we used our new cross-validation strategy with the experimental data, comparing models wherein the transitions matrix is either defined separately for each drug concentration or shared. This supported our decision because sharing the transitions matrix had no impact on the ability of the model to predict left-out observations (Fig. S16).

We have noted these issues in the results and discussion (lines 290–305 and 409–427).

Moreover, fitting a single distribution across multiple concentrations might not work. This could be addressed through synthetic simulations or by analyzing the concentrations individually but is at least [worthy] of discussion.

"We enforced a unidirectional phenotypic shift with drug concentration in AU565 cells, reflecting an expectation of a dose-response effect on cell phenotype within each state." I could not find a description of how this was implemented, but this sounds like a tricky operation that should really see some validation on synthetic data etc.

This could have been described more clearly. We have expanded our description of this method (lines 617–637) and repeated some of these points below.

This model constraint only influences the estimation of cell lifetimes during the maximization step. There, this constraint is implemented as a linear inequality constraint on the scaling parameter of the lifetimes' distribution. Fitting a Gamma distribution is known to be a convex optimization problem, meaning that the globally optimal solution can be derived from fitting the likelihood. Linear equality constraints within any convex optimization problem still result in a convex optimization problem. For these two reasons, we are confident that the optimization leads to the globally optimal result. As validation of this expectation, we ran the distribution fitting step with thousands of random starting points and observed that we always obtained the exact same solution. Therefore, we are extremely confident that this constraint in no way limits the model's ability to effectively fit the data.

As raised by other reviewers, I think it would be helpful to try and provide some more tangible interpretations of the results:

Are clusters 5 and 6 in the Lapatinib basically the same, cause by an artificial split? Alternately, if indeed 5 spends longer in S/G2 and then switches to 6 which spends less time, is this something that could be confirmed/followed up on, even without drugs?

Thank you for pointing out this unclear point in the model. Note that, due run-to-run variation in the model, the effect observed for clusters 5 and 6 is now switched to states 2 and 4 in the current results.

It is not possible for this to be an "artificial" split in the data. We think what you mean by an artificial split is that the emissions distribution is not a good representation of the observations for a single state, and so the data is split into two clusters. States 2 and 4, while rapidly transitioning, are not interchangeable—state 2 almost universally leads to state 4, and vice versa. An artificial split of the sort we think you are asking about would result in parallel paths in the transition diagram. For instance, state 1 would lead to both 2 and 4 with equal probability, and then both those states would lead back to a common downstream state. We do not see examples of this structure in our results. States 1 and 3 also cannot be examples of artificial splitting, because they are relatively "stable" states (mostly remaining the same state). This reflects that the parent strongly predicts the behavior of the daughter.

Would it be possible to provide [summary] results regarding how much knowledge of the initial state informs the future behavior of the cells, and how much better this does than a simple binning/k-means approach?

This is a great idea, though we struggled with exactly how best to address it at first. We have had similar discussions about how we might quantify the strength of inheritance within lineages in a model-agnostic way. For instance, we previously quantified the correlations between parent and daughter cells, cousin cells, etc., to try and trace how far inheritance effects extend. However, these results did not provide specific insights beyond the observation that related cells are more similar in their behavior.

To try and tackle this question more directly—about coming up with a summary of how much better state informs the behavior of cells—we determined a strategy using our model. Using our cross-validation strategy described above, we compared these results to an alternative model wherein we set the state transitions to all be equal. By doing this, the current state of the cell has no bearing on the state assigned to the daughter. Consequently, any knowledge gained by inheritance is lost. We have overlaid these results with the cross-validation results in Fig. S16. This clearly shows that the parent state provides important information to predict the behavior of daughters much more accurately.

Reviewer #5 (Remarks to the Author: Reproducibility):

6. A large portion of the methods was devoted to the tree-based Markov model, which I think has already been described by Durand et al. I think it would have been better to provide a simple summary of this method and focus on deviations from that work as adopted here.

Indeed, the method we used to implement the EM algorithm for the tHMM was the one introduced by Durand *et al.* We summarized some general concepts and provided a clearer notation to follow the algorithm, which the other reviewers appreciated and specifically requested be expanded. We extend the implementation proposed by Durand *et al.* by developing appropriate distributions for cell lifetimes, incorporating censored estimators, developing a modular and usable implementation for the community, and developing a cross-validation strategy. We have extensively reorganized and consolidated the entire methods text so that these advancements are explained separately from the general introduction (see the entire Materials and Methods section). Notably, the description of our advancements relies on the notation and ideas introduced in the description of elements borrowed from Durand *et al.*

Additionally, to make this work more reproducible, I would [have] liked to have seen further description of:

- 1. The pre-processing performed before the k-means analysis, to ensure this was not artificially handicapped.*

As described above, we have removed our use of k-means and instead compare our model to the analytically derived optimal clustering cutoff. We have added a description of this approach in lines 746–758.

2. *The censoring approach: The authors need to elaborate on the specific censored estimators/approach that were used, as censoring is part of the key novelty of this work.*

We have now added more explanation about how we account for data censoring in lines 696–705.

Is the "Random Index Accuracy" indicated in various graphs and in the rebuttal, the same as the Rand Index for clustering? If so, this is confusing and incorrect naming. I believe the Rand index derives its name from the statistician William M. Rand and is not a contraction of the word "Random". The standard Rand index does not address randomness, but if using the "Adjusted Rand Index" which does, this should be clarified.

Thank you for this explanation. We are indeed using the adjusted index developed by William M. Rand. We have now changed "Random Index Accuracy" to "Adjusted Rand Index Accuracy" throughout the manuscript.

We thank the reviewers for this third round of feedback. Please find a point-by-point response to each comment included below.

Reviewer #5 (Remarks to the Author: Overall significance):

I appreciate the effort the authors took to respond to my concerns. They have all been addressed adequately.

One minor suggestion: While the authors use of the adjusted rand index interchangeable with accuracy is fair, it would be kinder to the reader if the connection were made more explicit when first used.

Thank you for pointing this out. We have added an explanation about the adjusted rand index, and the reason we used it instead of accuracy, on first use in the results and figure captions.